# Centrosome, the Newly Identified Passenger through Tunneling Nanotubes, Increases Binucleation and Proliferation Marker in Receiving Cells

**DOI:** 10.3390/ijms22189680

**Published:** 2021-09-07

**Authors:** Fatéméh Dubois, Ludovic Galas, Nicolas Elie, Frank Le Foll, Céline Bazille, Emmanuel Bergot, Guénaëlle Levallet

**Affiliations:** 1Normandie University, UNICAEN, CEA, CNRS, ISTCT/CERVOxy Group, GIP CYCERON, F-14000 Caen, France; fatemeh.dubois@unicaen.fr (F.D.); bazille-c@chu-caen.fr (C.B.); bergot-e@chu-caen.fr (E.B.); 2Department of Pathology, CHU de Caen, F-14000 Caen, France; 3Normandie University, UNIROUEN, INSERM, PRIMACEN, F-76000 Rouen, France; ludovic.galas@univ-rouen.fr; 4Normandie University, UNICAEN, SFR ICORE, Plateau CMABio3, F-14033 Caen, France; nicolas.elie@unicaen.fr; 5Normandie University, UNIHAVRE, UMR-I 02 INERIS-URCA-ULH SEBIO FR CNRS 3730 Scale, CEDEX, F-76063 Le Havre, France; Frank.lefoll@univ-lehavre.fr; 6Department of Pulmonology & Thoracic Oncology, CHU de Caen, F-14033 Caen, France

**Keywords:** tunneling nanotubes, centrosome, tumor development

## Abstract

Type 1 tunneling nanotubes (TNTs-1) are long, cytoplasmic protrusions containing actin, microtubules and intermediate filaments that provide a bi-directional road for the transport of various components between distant cells. TNT-1 formation is accompanied by dramatic cytoskeletal reorganization offering mechanical support for intercellular communication. Although the centrosome is the major microtubule nucleating center and also a signaling hub, the relationship between the centrosome and TNTs-1 is still unexplored. We provide here the first evidence of centrosome localization and orientation towards the TNTs-1 protrusion site, which is implicated in TNT-1 formation. We also envision a model whereby synchronized reorientation of the Golgi apparatus along with the centrosome towards TNTs-1 ensures effective polarized trafficking through TNTs-1. Furthermore, using immunohistochemistry and live imaging, we observed for the first time the movement of an extra centrosome within TNTs-1. In this regard, we hypothesize a novel role for TNTs-1 as a critical pathway serving to displace extra centrosomes and potentially to either protect malignant cells against aberrant centrosome amplification or contribute to altering cells in the tumor environment. Indeed, we have observed the increase in binucleation and proliferation markers in receiving cells. The fact that the centrosome can be both as the base and the user of TNTs-1 offers new perspectives and new opportunities to follow in order to improve our knowledge of the pathophysiological mechanisms under TNT control.

## 1. Introduction

Since their discovery by Rustom and colleagues in 2004 [1], Tunneling nanotubes (TNTs) represent a direct mode of intercellular communication that are gaining widespread importance in our understanding of processes involved in health and disease [2,3,4]. During TNT formation, the cell architecture undergoes a dramatic change to form long cytoplasmic extensions, which hover freely above the substratum and reach a distance up to 100 μm [5,6]. While actin filaments are presents in all TNTs, the microtubules (MTs) are considered as the main component of the thicker subtype (type 1 TNT, hereafter referred to as TNTs-1 for simplicity) [7,8], and serve as tracks for direct exchanges of various components including organelles (e.g., mitochondria, lysosomes) between the cytoplasm of non-adjacent cells [9,10].

Microtubule nucleation and anchoring are mainly under centrosome control [11]. Indeed, in most epithelial cells, the centrosome moves from the perinuclear region towards the cell membrane to nucleate the axoneme of the ciliary projection [12]. Similarly, the reorientation of the centrosome toward the leading edge of motile cells results in the asymmetric distribution of the MTs as well as the establishment of front–rear cell polarity and oriented trafficking [13,14]. In neuronal cells, the centrosome position has an important role in axial elongation [15]. In dividing cells, the centrosome facilitates the assembly of astral MTs and dictates the position and orientation of the cleavage furrow during cytokinesis [16]. However, the involvement of the centrosome in the formation of TNTs-1 is still unexplored. It is therefore of interest to understand whether centrosome reorientation occurs during TNT-1 establishment to impact the subsequent MT reorganization. 

Furthermore, given the increasing evidence indicating a role of TNTs in tumor progression [3,4,10], some fundamental issues, such as the involvement of TNTs-1 in centrosome abnormalities, have to be addressed. As a conspicuous feature of many tumor cells, centrosome amplification can translate to malignant transformation and predicts poor patient prognosis [17]. However, the question of how abnormal cells with supernumerary centrosomes could survive has not been conclusively resolved. Herein, we induced concomitant centrosome amplification and TNT-1 formation in human epithelial bronchial cell lines and investigated whether TNT-1 formation could provide benefits to the cells with an extra centrosome to avoid genetic instability [18,19]. Our results provide the first evidence of movement of centrosomes within TNTs-1, and consequently lay the foundation for an exciting new research direction for improving our understanding of TNTs formation and their role in carcinogenesis.

## 2. Results and Discussion

### 2.1. The Centrosome Polarizes towards TNT-1 Protrusion Sites

Owing to the pivotal role of centrosomes in MT nucleation, we first investigated centrosome localization in TNTs-1-formed cells by using either anti-γ-tubulin and/or anti-pericentrin, which detect the protein matrix around the centrosome, named pericentriolar material [20]. As determined by immunofluorescence analysis of the human bronchial epithelial cells (HBEC-3), in 70% of cells that form TNTs-1, the centrosome dissociates from its tight association with the nuclear envelope (Figure 1A–D, yellow arrows) and becomes oriented towards the leading edge where the TNTs-1 forms or protrudes (Figure 1A–D, white arrows). To derive more reliable conclusions, these results were repeated and verified not only in three other different lung epithelial cell lines (H1299, A549, BEAS-2B) (Appendix A), but also in paraffin-embedded tissue sections of patients with non-small cell lung cancer (NSCLC) (Figure 1E).

Next, with regard to the well-established role of centrosome polarization in the asymmetric distribution of stable MTs [11], we hypothesize that a close apposition of the centrosome to the cell cortex probably guided the directionality of microtubules growth toward the site of TNT-1 protrusion. In fact, the same organization is found not only in migrating cells [13], but also during axon or cilia outgrowth, where the centrosome can settle at some distance apart from nucleus and polarize facing the cell extension site, in order to facilitate MT elongation [12,15]. To test this hypothesis and determine whether centrosomes are required for TNT formation, we treated the cells with Centrinone (LCR-263), a selective and reversible inhibitor of polo-like kinase 4 (PLK4), which causes centrosome depletion and arrests cells in a senescence-like G1 state [21]. Importantly, the function of the centrosome was shown to be critical for TNT-1 extension, as Centrinone treatment decreased TNT formation by 85% in HBEC-3 cells (Figure 1F,G).

However, centrosome displacement not only involves the reorganization of microtubules but also implies a reorientation of cell function and polarity through intimate structural and functional association with the Golgi apparatus and endocytic-recycling compartment [22,23,24]. Consistent with this scenario, immunofluorescence staining showed pericentrosomal polarization of the Golgi complex towards and within the TNT-1 structure (Figure 2A,B and Appendix A). In our previous experiments, intracellular vesicles, including recycling endosomes stained with anti-Rab11 [25] and lysosomes, stained with LysoTracker [26], were also observed within TNTs-1 in HBEC-3 cells [8]. These data support a model in which such geometrical organization brings the centrosome and Golgi complex close enough to the TNT-1 protrusion site in order to facilitate directional intercellular exchange of endocytic-recycling vesicles through TNTs-1 (Figure 2C). Furthermore, these data provide additional mechanistic context describing the role of centrosomes in the control of Golgi orientation [13].

Accordingly, it will be interesting to investigate whether the activity of centrosome, as a signaling platform [27], is also necessary for the recruitment and anchoring of additional molecules with specific functions in TNT-1 formation, such as the Cdc42 signaling molecule, which regulates both centrosome repositioning [28] and TNT-1 formation [29].

### 2.2. TNTs-1 as an Unexpected Road for Extra Centrosomes 

Our venture into studying the relationships between centrosomes and TNTs-1 prompted us to explore the pathological relevance of this connection in the context of centrosome amplification. Because of the MT nucleation capacity of extra centrosomes, we hypothesized that centrosome amplification would influence both the fidelity of chromosome transmission and tissue architecture including TNT-1 formation. For this purpose, we focused our attention on the scaffold protein RAS association domain family 1A (RASSF1A) because of its role in control of both centrosome number [30,31,32] and TNT-1 formation [8]. By using RNA interference (RNAi) treatment, we first took advantage of nontumorigenic, non-transformed HBEC-3 cells to avoid additional effects caused by cancer mutations [33]. 

Consistent with previous reports, RASSF1A depletion (Appendix A) leads to the development of nuclear atypia such as multi-nucleation (Appendix A) and micronuclei (Appendix A), as an indication of extra centrosome occurrence and mitotic spindle defects (Appendix A) [32,34]. Surprisingly, in almost 70% of the RASSF1A-depleted HBEC-3 cells that contain both extra centrosomes and TNTs-1, we observed a displacement of at least one of the centrosomes through TNTs-1 (Figure 3A,B). To obtain insight into the nanoscale details of the centrosome through TNTs-1, we used STED super-resolution microscopy, which provides an optical resolution well below the diffraction limit [7]. In STED images, the pericentrin staining was not only resolved into the ring shape instead of blobs under confocal microscopy, but also revealed that centrosomes clearly used the microtubule to travel through TNTs-1 (Figure 3C). This result was confirmed not only in lung epithelial A549 and H1299 tumorigenic cell lines with the hypermethylated RASSF1 gene (Appendix A), but also remarkably in paraffin-embedded tissue sections of the patients with NSCLC (Figure 3D). To determine whether these observations also hold true in live cells, HBEC-3 cells were subjected to transfection with vector encoding EGFP-centrin. Time-lapse imaging further confirmed the presence and movement of centrosome within TNTs-1 between interconnected cells (Figure 3E, Appendix A). To our knowledge, and except in the case of cell division, this is the first demonstration of the displacement of the centrosome from cell to cell.

This fascinating observation raised two main questions about the functional consequence of such localization within TNTs-1. In fact, numerous studies have showed that, despite high levels of centrosome amplification in low-grade lesions of the most human cancers [35], cells have various coping mechanisms to escape death caused by multipolar mitoses and still divide successfully [17,36]. One of these, certainly the best characterized mechanism, is the coalescence of supernumerary centrosomes into two functional spindle poles (Appendix A) [18,37,38].

According to these findings, we first asked whether the localization of extra-centrosome within TNTs-1 can account for an additional process to limit the detrimental consequence of centrosome amplification and to protect cells from genomic instability and transformation. We tested this hypothesis by performing co-culture experiments. To distinguish both cell populations, RASSF1A depletion was performed in labeled HBEC-3^Cherry^ cell line, while the normal HBEC-3 cell line was used for siNeg transfection. These populations were then cultured together in a 1:1 ratio for 48 h. Of importance and in agreement with our prediction, transport of extra centrosomes via TNTs-1 was also confirmed in coculture system from RASSF1A-depleted HBEC-3^Cherry^ cells toward control siNeg HBEC-3 cells (Figure 4A). Additionally, and in line with the failure of proper DNA segregation, we also noticed in some cases the localization of micronuclei through TNTs-1 (Figure 4B).

Another appealing idea is the possibility that TNTs-1 assist in the dissemination of extra centrosomes to manipulate the surrounding cells and participate in creating a highly heterogenous microenvironment. In support of this possibility, it has been reported that the transfer of genetic materials such as mRNA or microRNAs via TNTs could induce or repress the transcription of genes implicated in cancer cell motility or even enhance the transformation of normal neighbor cells [39,40]. Concordantly, the active transfer of mitochondria from normal cells to tumor cells with dysfunctional mitochondria, is able to restore normal cell respiration [3,41]. 

To address this idea, we conducted another series of complementary co-culture experiments. This time either non-treated HBEC-3^Cherry^ cells or RASSF1A-depleted HBEC-3^Cherry^ cells were (co-) transfected with GFP-centrin plasmid and then co-cultured with either siNeg-transfected HBEC-3 or non-treated A549 cell lines for 24 h. After fixation, we focused on the HBEC-3 and/or A549 cells that received GFP-centrin without being transfected. We hypothesize that using TNTs-1 is the unique conduit for intercellular transfer of centrosomes (identified by GFP-centrin) from donor HBEC-3^Cherry^ cells into receiving cells (without cherry signal) since the average size of centrosome matrix (in vertebrates) is 1.6 ± 0.5 μm^2^ [42,43,44] and among communication tools, only TNT-1 reach a thicknesses of over 700 nm sufficient to transfer of whole centrosomes from donor cells into receiving cells [45,46]. Indeed, gap junctional pores range in size from 11 to 24 Å in diameter [47] and extracellular vesicles, from nanometer-size exosomes (30–100 nm) to submicron-size microparticles (100–1000 nm) [48,49].

As expected, due to the increase in TNT-1 formation after RASSF1A depletion [8], the number of control cells that were co-cultured with RASSF1A-depleted HBEC-3^Cherry^ cells and presented GFP-centrin signals was significantly higher than control cells co-cultured with non-treated HBEC-3^Cherry^ cells (Figure 4C and Appendix A). Remarkably, our results show that the number of binucleated cells was increased in GFP-centrin acceptor cells compared to control cells that did not receive any centrosomes (Figure 4D,E). Accordingly, another experiment revealed that the presence of GFP-centrin in normal cells caused a significant increase in proliferation identified by Ki67 staining (Figure 4F,G). Altogether, these exciting findings fit a model in which TNTs-1 serves as a critical pathway to transfer extra centrosomes from malignant cells and alter healthy cells in the tumor environment. In future experiments, it will also be interesting to assess whether centrosome transfer leads to other features of malignant transformation, such as invasive protrusions in acceptor cells [17].

Last but not least, it should keep in mind that intercellular communication through TNTs-1 is a two-way process and can not only promote but also inhibit tumor growth, depending on the transferred materials and their effect on cell activation status. In this regard, the present observations can form the basis for new therapeutic strategies that target cells carrying extra centrosomes.

Collectively, in this short report, we suggest that the location of the centrosome not only predicts the site of TNT-1 formation but also may facilitate intercellular trafficking through the regulation of both microtubule formation and Golgi orientation. Furthermore, except for the case of dividing cells, our results provide the very first strong assumption of centrosome displacement from cell to cell even if we failed to show the actual transfer of centrosome from the donor cell to the recipient cell directly. Accordingly, the displacement of extra centrosomes through TNTs-1 might protect cells from the deleterious impact of centrosome amplification, which can translate to malignant transformation and poor patient prognosis. Clearly, more studies are needed to substantiate our findings and their pathological implications during the carcinogenesis. However, herein, we undeniably lay the basis for a new research direction for improving our understanding of TNTs-1 roles in tumorigenesis processes.

## 3. Materials and Methods

### 3.1. Cell Culture and Transfection

Isogenic HBEC-3 (and HBEC-3-cherry) bronchial epithelial cell lines were a generous gift from Dr. White (UT Southwestern Medical Center, Dallas, TX, USA) and were cultured in keratinocyte serum-free medium (KFSM) complemented with 0.2 ng/mL of human recombinant epidermal growth factor (EGF) and 25 μg/mL of bovine pituitary extract (BPE) supplements (Thermo Fisher Scientific, Rockford, IL, USA). The other tumorigenic epithelial cell lines, BEAS-2B, H1299 and A549, were obtained from ATCC and were cultured in high-glucose Dulbecco’s modified essential medium (DMEM; Gibco, Waltham, MA, USA) with 10% fetal bovine serum and 2 mM of L-glutamine. All the mediums were also complemented with 10% (*v*/*v*) heat-inactivated fetal bovine serum, 100 U/mL penicillin, and 100 µg/mL streptomycin (Gibco). The cultures were incubated at 37 °C in a humidified atmosphere with 5% CO_2_. Cells were transfected using Lipofectamine RNAiMAX (Invitrogen, Carlsbad, CA, USA) at 30% confluence according to the manufacturer’s instructions and analyzed 72 h after treatment as described in our previous studies [8,33]. The following RNAi oligonucleotides from Eurogentec^®^ were used: RASSF1A: si1: 5′-GACCUCUGUGGCGACUUCATT-3′ [50] and non-targeting control RNAi (siNeg) from Dharmacon using a concentration of 0.1 µM for each oligonucleotide. Transient transfection with plasmids encoding centrin-green fluorescent protein (EGFP) (Addgene^®^, Teddington, UK) were performed using Lipofectamine RNAiMAX (Invitrogen, Carlsbad, CA, USA) following the manufacturer’s instructions and analyzed 24 h after transfection. To investigate whether TNT formation was dependent on centrosomes, Centrinone [LCR-263] (MCE) was added to the cell culture for seven days at the final concentration of 100 nM as previously described [20].

### 3.2. Preparation of RNA and RT-qPCR

Extraction of total RNA from treated and untreated cells was performed using the Illustra RNAspin mini^®^ column (GE Healthcare, Bio-Sciences, Pittsburgh, PA, USA) according to the manufacturer’s instructions. Total RNA was treated with DNAse I (Invitrogen, Carlsbad, CA, USA) to remove contaminating genomic DNA. The RNA concentrations were determined with spectrophotometer Nanodrop^®^ 2000c. Total RNA (250 ng) was reverse-transcribed with random primers and 200 IU M-MLV reverse transcriptase at 37 °C for 90 min, followed by 5 min of dissociation at 70 °C with Mastercycler Eppendorf^®^. The resulting cDNAs were diluted (1/10) and used as templates. PCR performed in Mx3005P QPCR system (Agilent Technology, Santa Clara, CA, USA) with 5 pmol of RASSF1 primer sets as follows: Forward (F): GGG GTC GTC CGC AAA GGC C and Reverse (R): GGG TGG CTT CTT GCT GGA GG. Actin was used as an internal control. Forward (F): CAA CCG TGA AAA GAT GAC CCA G and Reverse (R): ATG GGC ACAGTG TGG GTG AC.

### 3.3. Immunofluorescence and Time-Lapse Video Microscopy

For immunofluorescent staining, the cells were fixed 72 h after transfection. For co-cultured experiments, at 48 h after transfection with either siNeg or siRASSF1A, HBEC-3 and HBEC-3-cherry cells were trypsinized and seeded together at 60% confluence into 24-well plates containing coverslips coated with poly-L-lysine (Sigma; St Louis, MO, USA). Cells were allowed to grow for an additional 24 h. The cells were washed with PBS, fixed with paraformaldehyde (4%), and permeabilized with cold methanol for 10 min. The slides were treated with 1% BSA in PBS for one hour. The cells were incubated with the following antibodies for 1 h at room temperature: rabbit polyclonal pericentrin (Abcam, Cambridge, UK, 1:400), mouse-monoclonal Golgin-97 (Cell Signaling Technology, Danvers, MA, USA, 1:400), rabbit-polyclonal anti-ki67 (Abcam, Cambridge, UK, 1:100), mouse-monoclonal gamma tubulin (Abcam, Cambridge, UK, 1:200) and a mouse monoclonal anti-α-tubulin antibody (Molecular Probes; Eugene, OR, USA, 1:300). After washing with PBS, the Alexa Fluor 488-conjugated anti-rabbit and 643-conjugated anti-mouse secondary antibodies (Molecular Probes) were used, and DNA was counterstained with DAPI (Ultracruz Mounting Medium, Santa Cruz Biotechnology, Dallas, TX, USA). Images were obtained with an Olympus™ FluoView FV1000 laser scanning confocal microscope. Color images were processed using ImageJ.

To follow centrosome displacement through TNTs in live cells, EGFP-centrin transfected HBEC-3 cells were cultured in glass bottom cell culture dishes and monitored by video microscopy in a humidified, CO_2_-equilibrated chamber (Incubator i8, Singapore) using an inverted microscope (Leica DMi8, Wetzlar, Germany) equipped with a motorized stage controlled by Metamorph 7.8.13.0 software. The acquired images were then analyzed with the ImageJ software (version 1.50d). 

### 3.4. Tissue Samples and Immunohistochemical Staining

NSCLC tissue samples were from the Caen University Hospital and were obtained in accordance with the Declaration of Helsinki; all patients provided informed consent regarding the collection of tumor specimens and their molecular evaluation, as required by French laws, and the study was approved by the institutional ethics committee of Caen University Hospital, France (DC-2008-588). Tissue samples were formalin-fixed and embedded in paraffin. Hemalun and eosin-stained sections from original block were reviewed by a pathologist (CB) and used to select a representative area. Three micrometer sections were prepared from paraffin-embedded blocks and placed on superfrost plus slides. After antigen retrieval with pH 9.0 EDTA, immunohistochemistry was performed using a manual technique. After incubation overnight at 4 °C with primary antibody Pericentrin (Abcam, rabbit polyclonal, 1:400), detection was performed using Novolink detection kit (Leica^®^, Wetzlar, Germany) according to the manufacturer’s instructions. 

### 3.5. Quantification of the Centrosome and Golgi Complex Position in Cell

In each TNT-formed cell, the labeled centrosome and Golgi complex position were determined using a cell body centroid analysis, which divides the cell body into quadrants (perpendicular lines crossing the center of the nucleus divided the cell into four 90° quadrants) oriented toward the TNT protrusion site. The centrosome and Golgi were scored as “Front” if it were positioned within a 90° sector facing the TNT protrusion. Grids were aligned manually to the TNT axis, and quantification was obtained in a blinded fashion.

### 3.6. TNT Quantification

As there are no known specific TNT markers, we took into account three phenotypic criteria to avoid this confusion with similar-looking structures. As reviewed recently [9], the TNTs are thin and long cytoplasmic extensions that (i) do not touch the substrate, (ii) connect at least two cells, and (iii) are able to transfer the cargo or signal between the interconnected cells. Among the two classes of TNT, type 1 TNTs are thicker and contain not only actin but also microtubules and intermediate filaments, and type 2 TNTs are thin and contain only actin filaments [6]. To pursue this issue with functional experiments, we focused only on type 1 TNTs, because of their characterization for the transport of the large material (e.g., lysosomes, mitochondria) along microtubules [6,7], while type 2 TNTs, assimilated to the adherent’s junction, does not allow the exchange of biological material.

### 3.7. STED Imaging

Image acquisitions were performed with a 93× glycerol immersion objective (NA 1.30) through STED 3X imaging (TCS SP8; Leica Microsystems, Mannheim, Germany) with optimized parameters for Alexa 647 and Alexa 488 sequential detection. Within samples (zoom 6, pixel size = 20 nm), Alexa 647 and Alexa 488 were excited with the 650 and 499 nm wavelength of a supercontinuum laser, respectively. Depletion was obtained with 775 and 592 nm lasers for Alexa 647 and Alexa 488, respectively. Fluorescence (660–760 nm for Alexa 647 and 510–560 nm for Alexa 488) was collected with a hybrid detector and a pinhole for Airy 1.

The deconvolution of raw data from STED imaging was obtained through image processing with Huygens professional 4.5.1 software (SVI, Hilversum, The Netherlands). 

### 3.8. Statistical Analysis

Data are represented as the mean ± SEM of experiments performed independently at least three times. To determine statistical significance, Student’s unpaired *t* test was applied to all experiments. Statistical significance was set at *p* < 0.05.

## Figures and Tables

**Figure 1 ijms-22-09680-f001:**
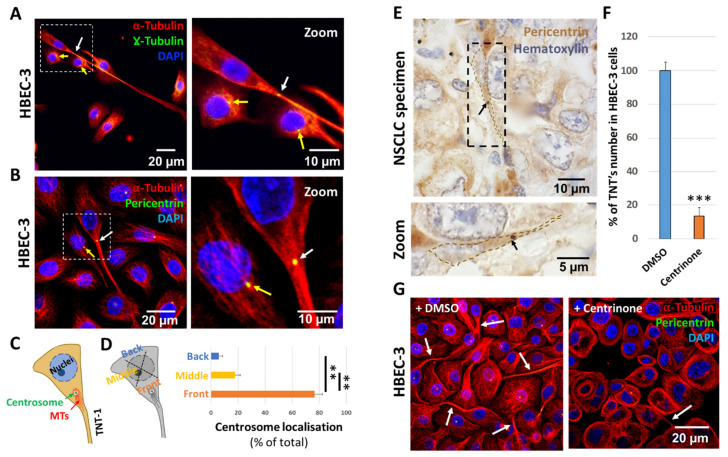
The centrosome polarize towards TNT-1 protrusion sites. (**A**,**B**) Representative images, and (**C**) illustration showing the localization of centrosome at the base of the formation of TNT-1. Dashed white squares indicate the regions used for the zoom. White arrows indicate centrosome polarization towards TNT-formed cells and yellow arrows represent normal centrosome position near nucleus. (**D**) Figurative illustration showing the method of centrosome quantification and quantification of centrosome orientation towards TNT’s-1 protrusion site. *n* > 200 cells. In each TNT-formed cell, the labeled centrosome position was determined using a cell body centroid analysis, which divides the cell body into quadrants. The centrosome was scored as “Front” if it was positioned within a 90° sector facing the TNT-1 protrusion as illustrated. (**E**) Representative immunohistochemistry staining of pericentrin in paraffin-embedded NSCLC tumor section. Dashed black squares indicate the regions used for the zoom. Black arrows show centrosome localization. (**F**) Quantification and (**G**) representative images of TNT-1 number in HBEC-3 cells treated with Centrinone (100 nM) for seven days. *n* > 200 cells. White arrows show centrosome localization. Data are represented as the mean ± SEM from three individual experiments. Statistical significance was calculated and *p* value are indicated by asterisks: ** *p* < 0.01 and *** *p* < 0.001.

**Figure 2 ijms-22-09680-f002:**
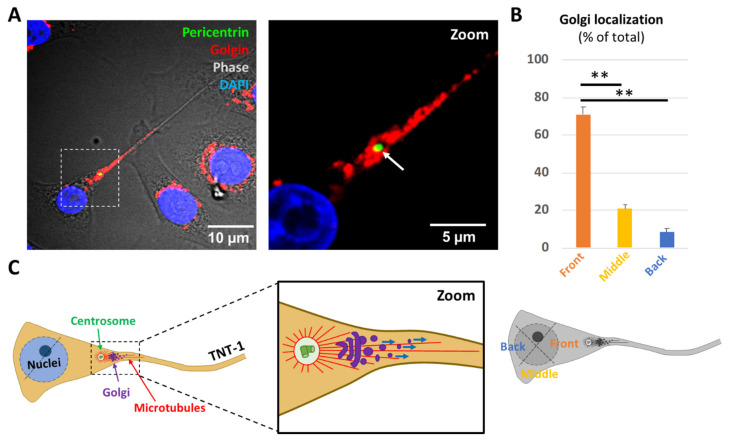
Pericentrosomal localization of the Golgi towards TNT’s-1 formation site facilitates intercellular trafficking. (**A**) Representative images (**B**) and quantification of pericentrosomal localization of the Golgi apparatus towards and within TNTs-1. *n* > 100 cells. In each TNT-formed cell, the labeled Golgi complex position was determined using a cell body centroid analysis, which divides the cell body into quadrants. The Golgi was scored as “Front” if it was positioned within a 90° sector facing the TNT-1 protrusion as illustrated. Dashed white square indicates the region used for the zoom. White arrows indicate centrosome polarization towards TNT-formed cells. (**C**) Figurative illustration of pericentrosomal localization of the Golgi apparatus and endocytic-recycling compartment towards and within TNTs-1. Boxed region used for the zoom. Arrows indicates centrosome, Golgi complex and microtubules position. Blue arrows mark bright vesicles that moved along the TNT-1. Data are represented as the mean ± SEM from at least three individual experiments. Statistical significance was calculated and *p* value are indicated by asterisks: ** *p* < 0.01.

**Figure 3 ijms-22-09680-f003:**
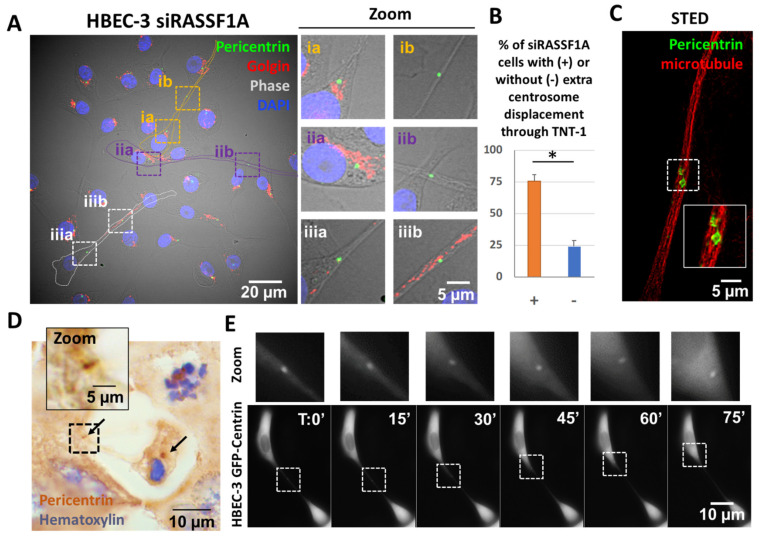
The presence and movement of centrosome within TNTs-1. (**A**) Representative images, with zoom focusing on three cells with TNT-1 (i, ii, iii), showing centrosome at the base of TNT-1 (ia, iia, iiia) or in the TNT (ib, iib, iii b). Dashed squares indicate the region used for the zoom. (**B**) Quantification of centrosome displacement through TNT-1. *n* > 70 cells. The cells were scored as (+) if extra centrosome was positioned within TNT-1. Data are represented as the mean ± SEM from at least three individual experiments. Statistical significance was calculated and *p* value are indicated by asterisks: * *p* < 0.05. (**C**) STED super-resolution microscopy reveals ring shape pericentrin staining using microtubule to move through TNT. Dashed white square indicates the region used for the zoom (**D**) Representative immunohistochemistry staining of pericentrin in paraffin-embedded NSCLC tumor section. White arrows indicate centrosome position. Dashed black square indicates the region used for the zoom. Black arrows show centrosome localization. (**E**) Montage of time-lapse imaging of EGFP-centrin transfected HBEC-3 cells showing centrosome movement through TNT-1. The Centrin2-RFP signal is presented as green, but the brightness and contrast were adjusted to identify the Centrin signal in the black and white background. Dashed white square indicates the region used for the zoom.

**Figure 4 ijms-22-09680-f004:**
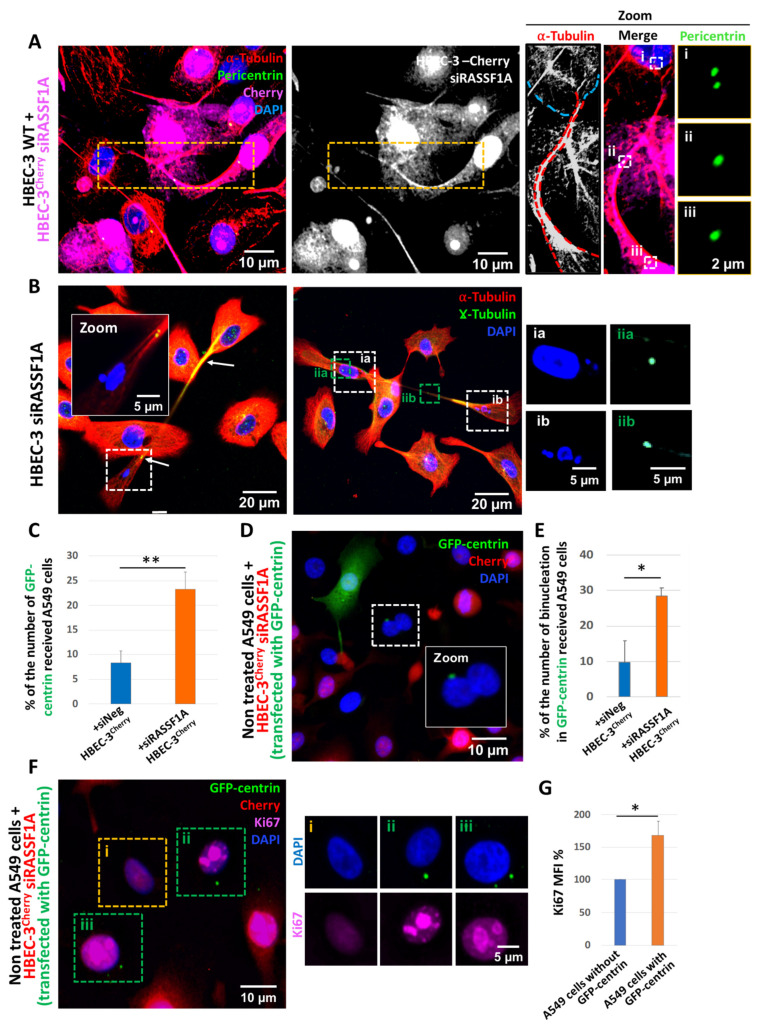
Centrosome displacement through TNTs-1 contributes to alter cells in the environment. (**A**) Representative images of displacement of extra-centrosome through TNTs-1 from HBEC-3-Cherry cells transfected with siRASSF1A to HBEC-3 siNeg-transfected. Dashed orange rectangle indicates the region used for the first zoom, while small dashed white squares indicate the regions used for the second zoom. (**B**) Representative images showing micronuclei displacement through TNT-1. “Green” boxs show centrosome, “White” box show micronuclei. For the sake of clarity, the zoom pictures have been captured in another z-planes (**C**) Quantification of the number of GFP-centrin received A549 cells in co-culture with either siNeg- or siRASSF1A- transfected HBEC-3Cherry. (**D**) Representative images and (**E**) quantification of the number of binucleated A549 cells in co-culture with either siNeg- or siRASSF1A- transfected HBEC-3Cherry. Dashed white square indicates the region used for the zoom. (**F**) Representative images of Ki-67 staining in co-culture of Non treated A549 cells + HBEC-3Cherry siRASSF1A (transfected with GFP-centrin) to monitor cell proliferation. Boxed regions used for the zoom. “Green” boxs show the A549 cells with GFP-centrin, “Orange” box show the A549 cells without GFP-centrin. (**G**) Quantification of ki-67 MFI per cell. Data are represented as the mean ± SEM from at least three individual experiments. *n* > 200 cells. Statistical significance was calculated and *p* value are indicated by asterisks: * *p* < 0.05, ** *p* < 0.01.

## Data Availability

The original data can be found in the following address: centrosome.transfer@gmail.com Password: Centrosome 2020.

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
