# Peer review of "Centrosome, the Newly Identified Passenger through Tunneling Nanotubes, Increases Binucleation and Proliferation Marker in Receiving Cells"

_ijms, 2021, doi:10.3390/ijms22189680_

Round 1

Reviewer 1 Report

The goal of this short report is to describe the intercellular transfer of centrosomes between HBEC cells via tunneling nanotubes (TNTs).  This is of great interest in the field since it would add to the list of distinct cellular compartments that can be transferred via TNT like structures and could have an important role in cancer.

While very interesting, the authors did not convincingly demonstrated that the centrosomes are in fact moving from one cell to another via TNTs (which is the entire premise of this report).  There are experimental controls and details of the data acquisition and analysis missing that need to be addressed.

General comments:

  1. The authors called the structures they are looking at TNTs.  These are not TNTs as described by the field.  TNTs are F-actin tubular structures that allow for the intercellular transfer of a number of organelles and signals.  The structures described in this articles are much larger and contain microtubules, thus should not be referred to simply as TNTs but more specifically as microtubule-containing TNTs or MT-TNTs (for references see Wand and Gerdes, Cell death and differentiation, vol 22, 2015 or more recently, Han and Wang, IJMS vol. 22 2021).  The different types of TNT like structures should be briefly explained in the introduction.
  2. The authors appear to use different words such as “polarizes towards”,  “displacements”, “displacement through”, “road to evacuate” and “transfer” interchangeably.  However these words are not all synonyms and should not be used appropriately.

3)   Overall, the evidence for actual transfer of centrosomes through TNTs is not conclusive.  While it is clear that the centrosomes are “polarize toward” or are “displaced” and can be found at the base or within MT-TNTs, the authors did not conclusively demonstrated transfer.

4)   In addition, there is no control for the possibility of centrosome transfer other than by TNTs either.  For instance in l. 175, the authors stated “ We assume that using TNT is the unique conduit for intercellular transfer of centrosome…’. Why not actually test this assumption by adding a microtubule destabilizing drug or centrinone to reduce TNT formation with the co-cultured cells to show that GFP-pericentrin is no longer observed in the receiving cells?  

Specific comments:

Fig. 1:

(D) 200 cells were quantified in how many independent experiments?

(F) how many independent experiments are used and how many cells were counted?

Fig. 2:

(B) Quantification of 50 cells is low.  How many independent experiments were acquired?  50 cells should be the minimum for 1 independent experiment and a minimum of 3 independent experiments should be acquired.

Fig. 3:

(B) Quantification of centrosome displacement through TNTs.  The authors show the “% of siRASSF1 cells with or without extra centrosome displacement through TNTs” .  Here there is an important control missing.  The authors need to add a “scramble RNA control” to show the effect of siRNA and the difference in the centrosome displacement in siRASSF1 cells.  In addition, similar to the previous figures, the authors need to add how many independent experiments were done and n=40 is once again very low.

(E) and Movie 1:  This example does not show the transfer of centrosome through TNTs.  The movie/time lapse starts with the centrosome already in the tube and then show movement towards the top cell.  What is shown here is the movement of centrosome in the tube, not transfer.  In order to show transfer the centrosome would have to be shown moving from the bottom cell to the top cell.  This is not the case.  The centrosome could have come from the top cell, move ~ 10 um down within the tube and then go back toward the center of the cell it originally came from.

Fig. 4:

(A) The tubular structure from the mCherry cell doesn’t appear to touch the neg cells on the left where the other points to the 2 centrosomes in (ii).  It would help to have the phase images overlaid with the fluorescent images (like in Fig. 2 and 3).

In addition, there appears to be an mCherry cells on top of the negative cell (right outside of the highlighted yellow box ).  It is very difficult to see the outline of that cell and where its limits are.  Could the Pericentrin come from that cell?  A phase image could help better visualizing each individual cell and the cellular protrusion.

(C), (E) and (G): There is no information on how many independent experiments were acquired and how many cells were counted in order to generate these plots.  This is important in order to determine the significance of the data.

Sup. Fig 4:

In general, it is difficult to see the TNTs actually bound to the other cells.  In Sup. Fig 4, the TNTs observed appear not to connect to the “receiving cells”.  As stated for Fig. 4., it would help to have the phase images overlaid with the fluorescent images (like in Sup. Fig 5).  

Sup. Fig. 4 and 5: 

These Figures do not show transfer but simply the presence of centrosomes within the long cellular protrusions.  Transfer means movement from cell A to B.  None of these still images show that.

Sup. Fig 6:

It doesn’t show TNT between the transfected cell and the non-treated A549 cell.

Author Response

We would like to thank the reviewer #1 for his/her interests and valuable comments on the manuscript. As suggested, our article has undergone English language editing by MDPI. The text has been checked for correct use of grammar and common technical terms, and edited to a level suitable for reporting research in a scholarly journal.

General comments :

  1. The authors called the structures they are looking at TNTs.  These are not TNTs as described by the field.  TNTs are F-actin tubular structures that allow for the intercellular transfer of a number of organelles and signals.  The structures described in this article are much larger and contain microtubules, thus should not be referred to simply as TNTs but more specifically as microtubule-containing TNTs or MT-TNTs (for references see Wand and Gerdes, Cell death and differentiation, vol 22, 2015 or more recently, Han and Wang, IJMS vol. 22 2021).  The different types of TNT like structures should be briefly explained in the introduction.

As requested, the characteristic of TNTs have been explained in the introduction. According the characterization established by Bénard and co-workers, in 2015, there are two classes of TNT:

  • TNT type 1 which are thicker and contain not only actin but also microtubules and intermediate filaments and
  • TNT type 2 which are the thin ones and contain only actin filaments.

The data in the literature suggest that these two types of TNT could have different functions, as large material (e.g., lysosomes, mitochondria) can only travel between cells via TNT type 1 on microtubules (Onfelt et al., 2006; Bénard et al., 2015; Wang & Gerdes, 2015). For these reasons and to pursue with functional experiments, we have focused only on type 1 TNT.

As requested by reviewer and to avoid any confusion for the reader, we have modified the text of our manuscript and now specify “TNTs” when talking about all TNTs, and, TNTs-1 when talking about TNTs type 1.

  1. The authors appear to use different words such as “polarizes towards”,  “displacements”, “displacement through”, “road to evacuate” and “transfer” interchangeably.  However these words are not all synonyms and should not be used appropriately.

We thank the reviewer for pointing this out. As requested, we tried to homogenize the use of these words.

3)   Overall, the evidence for actual transfer of centrosomes through TNTs is not conclusive.  While it is clear that the centrosomes are “polarize toward” or are “displaced” and can be found at the base or within MT-TNTs, the authors did not conclusively demonstrated transfer.

We thank the reviewer for raising this critical issue. According to the reviewer’s suggestion, we have replaced the word “ transfer” with either “displacement” or “displace”

4)   In addition, there is no control for the possibility of centrosome transfer other than by TNTs either.  For instance in l. 175, the authors stated “ We assume that using TNT is the unique conduit for intercellular transfer of centrosome…’. Why not actually test this assumption by adding a microtubule destabilizing drug or centrinone to reduce TNT formation with the co-cultured cells to show that GFP-pericentrin is no longer observed in the receiving cells?  

We thank the reviewer for raising this important topic. We assume that using TNT is the unique conduit for intercellular transfer of centrosome because according to the literature, except in the case of cell division, the centrosome transfer from cell to cell, was never observed in any other context. However, to respond to the reviewer’s concern, we can’t realize these experiments due to technical impossibility. GFP-centrin transfection and co-culture experiments are realized after 24h but the centrinone treatment is effective after 7 days. Additionally, the Trypsination of treated cells with either centrinone or microtubule destabilizing drug cause the death of majority of cells prior to co-culture.

Specific comments:

Fig. 1:

(D) 200 cells were quantified in how many independent experiments?

The detail of each experiment has been added in the legends. Briefly, during three individual experiments (n=3), we have quantified the number of TNTs-1 in approximately 200 cells based on the characteristic discussed earlier.

(F) how many independent experiments are used and how many cells were counted?

The detail of each experiment has been added in the legends. Briefly, during three individual experiments, we have quantified the number of TNTs-1 in at least 200 cells based on the characteristic discussed earlier.

Fig. 2:

(B) Quantification of 50 cells is low.  How many independent experiments were acquired?  50 cells should be the minimum for 1 independent experiment and a minimum of 3 independent experiments should be acquired.

We thank the reviewer for pointing this out. In our revised manuscript, we have quantified the number of Golgi localization in another 50 TNTs-1 formed cells. Data are represented as the mean ± SEM from at least three individual experiments.

Fig. 3:

(B) Quantification of centrosome displacement through TNTs.  The authors show the “% of siRASSF1 cells with or without extra centrosome displacement through TNTs” .  Here there is an important control missing.  The authors need to add a “scramble RNA control” to show the effect of siRNA and the difference in the centrosome displacement in siRASSF1 cells.  In addition, similar to the previous figures, the authors need to add how many independent experiments were done and n=40 is once again very low.

We think there may have been some confusion here. The control cells don’t have any extra centrosome. In our revised manuscript, we have quantified the displacement of centrosome in another experiment to increase the number of cells. Data are represented as the mean ± SEM from at least three individual experiments.

(E) and Movie 1:  This example does not show the transfer of centrosome through TNTs.  The movie/time lapse starts with the centrosome already in the tube and then show movement towards the top cell.  What is shown here is the movement of centrosome in the tube, not transfer.  In order to show transfer the centrosome would have to be shown moving from the bottom cell to the top cell.  This is not the case.  The centrosome could have come from the top cell, move ~ 10 um down within the tube and then go back toward the center of the cell it originally came from.

We thank the reviewer for raising this critical issue. According to the reviewer’s suggestion, we have replaced the word “ transfer” with “displacement” .

However, it is worth to mention that this experiment presents many technical difficulties and challenges:  TNTs are very fragile, sensitive to the light and transient structures with an average survival time ranging from a few to tens of minutes (Sowinski et al., 2008 Bukoreshtliev et al., 2009). In addition, the movement of GFP-centrin is very difficult to visualize due to numerous issues:

  • unlike other organelles, which may number in the thousands per cell, centrosome typically occur in only one or two copies.
  • centrosomes are very small structures (2 microns) and they become out of focus very easily
  • and because of their small size, the strength of fluorescence signal reduces quickly and irremediably.

Fig. 4:

(A) The tubular structure from the mCherry cell doesn’t appear to touch the neg cells on the left where the other points to the 2 centrosomes in (ii).  It would help to have the phase images overlaid with the fluorescent images (like in Fig. 2 and 3). In addition, there appears to be an mCherry cells on top of the negative cell (right outside of the highlighted yellow box ).  It is very difficult to see the outline of that cell and where its limits are.  Could the Pericentrin come from that cell?  A phase image could help better visualizing each individual cell and the cellular protrusion.

The TNTs are considered not attached to the substrate as they hover freely in medium and we can observe the bodies of cells and the middle of TNTs in two different optical sections and not with the same focus through microscope. In this regard, TNTs are even capable of passing above the other attached cells. The reviewer is right, in this image, in the continuity of mcherry, we can observe the cell-body of the neg cells (please see MT-staining) which is without mcherry fluorophores. We apologize from reviewer because we don’t have the phase images corresponding.

(C), (E) and (G): There is no information on how many independent experiments were acquired and how many cells were counted in order to generate these plots.  This is important in order to determine the significance of the data.

We apologize for our carelessness and thank the reviewer for pointing this out. We have added the details in the legend. Briefly, data are represented as the mean ± SEM from at least three individual experiments. n>200 cells per individual experiment. Statistical significance was calculated and p value are indicated by asterisk: *p<0.05.

Sup. Fig 4:

In general, it is difficult to see the TNTs actually bound to the other cells.  In Sup. Fig 4, the TNTs observed appear not to connect to the “receiving cells”.  As stated for Fig. 4., it would help to have the phase images overlaid with the fluorescent images (like in Sup. Fig 5).  

The TNTs are fragile structures which may break during fixation. This is why in some images we cannot see the point of attachment to the other cells. But as requested by reviewer, we have replaced the word “transfer” with “displacement” to focus on the presence of centrosome through TNTs and avoid any confusion for the reader in these images. We apologize from reviewer because we don’t have the phase images corresponding.

Sup. Fig. 4 and 5: 

These Figures do not show transfer but simply the presence of centrosomes within the long cellular protrusions.  Transfer means movement from cell A to B.  None of these still images show that.

We thank the reviewer for raising this critical issue. According to the reviewer’s suggestion, we have replaced the word “ transfer” with “displacement” .

Sup. Fig 6:

It doesn’t show TNT between the transfected cell and the non-treated A549 cell.

The reviewer is right, there is no TNT between these cells, but we assume that the transfer has been occurred trough TNTs before the time of image acquisition as there is no others possibilities for centrosome to move from GFP transfected cells to control cells.

Reviewer 2 Report

The manuscript by Dubois and colleagues provides a fine description of the dynamics of Tunneling Nanotubes in human cultured cells. They show an intriguing correlation between centrosome positioning and Tunneling Nanotube organization. The Authors also identified a new and unusual role of the Tunneling Nanotubes as main actors to remove extra-centrosome and thus protect tumour cells against aberrant centrosome amplification. This study is quite interesting and might be very useful for people working on cell-to-cell communication. However, before publication the authors should address the following minor criticisms.

l.47 “transfer of centrosome” “apical-basal array of microtubules” the centriole moves toward the cell membrane to nucleate the axoneme of the ciliary projection.

l.57. “the increasing evidences indicating a role of TNTs in tumor progression” please provide a reference

l.73. “amorphous protein matrix around the centrosome named pericentriolar” Several observations suggest that the pericentriolar material is not amorphous. See Mennella et al., 2014.

Fig. 1. I have a stupid question and my personal curiosity: how the AA can be sure that the structures they indicate as TNTs are really TNTs and not elongate cytoplasmic protrusions as frequently are found in cultured cells? Are these protrusions fusing with other cells? If it is so, could the AA add a detail to show this point?

l.92. “centrosome was shown to be critical for TNT extension as Centrinone treatment decreased TNT formation by 85% 93 in HBEC-3 cells”. This finding is very interesting. However, can the AA exclude the possibility that surface protrusions are not formed because the cells are immotile?

l.151.  “TNTs can account for an additional process to limit the detrimental consequence of centrosome amplification and to protect cells from genomic instability and transformation” This is an interesting question. However, this transfer reduced the centrosomes within a cell, but increased the number of centrosomes in other cells. Thus, one cell avoids centrosome amplification, but other cells will have multiple centrosomes. I do not understand the advantage for the cell community. Please, explain.

  1. 186. “these exciting findings fit a model in which TNT serves as critical pathway to transfer extra-centrosome from malignant cells and alter healthy cells in the tumor environment”. Is it the answer to my previous question?

l.159. “in line with the failure of proper DNA segregation, we have also noticed in some case, the transfer of micronuclei through TNTs” I see in Fig 4 small spots at the basis of the cytoplasmic extensions, not through them. I think that this finding is not enough to affirm that micronuclei can move through TNT,

Fig. 4A. Alpha-tubulin staining is poor. Any way the localization of pericentrin is evident.

Fig. 4B. The shape of the nucleus and tubulin staining are different in low magnification and detail.  I do not understand the arrow pointing the elongated yellow projections: is it gamma-tubulin? If it is so, this finding is very unusual. Please explain.

l.237. anti-ki67. Is it mono or polyclonal?

The gamma-tubulin antibody is lacking from materials and methods section. Again, if the gamma-tubulin antibody is from Sigma (as usual), how the AA can detect both alfa and gamma tubulin? I think that this is a polyclonal antibody.

Author Response

We thank the reviewer for his/her enthusiastic support of the manuscript. Changes to the text are underlined and highlighted.

l.47 “transfer of centrosome” “apical-basal array of microtubules” the centriole moves toward the cell membrane to nucleate the axoneme of the ciliary projection.

The sentences have been modified.

l.57. “the increasing evidences indicating a role of TNTs in tumor progression” please provide a reference

The references have been added.

l.73. “amorphous protein matrix around the centrosome named pericentriolar” Several observations suggest that the pericentriolar material is not amorphous. See Mennella et al., 2014.

We thank reviewer for this remark. The sentence has been corrected.

Fig. 1. I have a stupid question and my personal curiosity: how the AA can be sure that the structures they indicate as TNTs are really TNTs and not elongate cytoplasmic protrusions as frequently are found in cultured cells? Are these protrusions fusing with other cells? If it is so, could the AA add a detail to show this point?

The TNTs are considered not attached to the substrate as they hover freely in medium and we can observe the bodies of cells and the middle of TNTs in two different optical sections and not with the same focus through microscope. To ensure that it is the case, we either observed TNT-1 (hereafter MT-TNT) in time laps (we thus can see that TNTs are even capable of passing above the other attached cells or observed TNTs on fixed cells with confocal microscope and not with an epifluorescence microscope to allow discrimination of cells extension touching the substrates from cell bridge (please see Dubois et al., 2018). However, to avoid any confusion for the reader, we have added more details in the M&M.

l.92. “centrosome was shown to be critical for TNT extension as Centrinone treatment decreased TNT formation by 85% 93 in HBEC-3 cells”. This finding is very interesting. However, can the AA exclude the possibility that surface protrusions are not formed because the cells are immotile?

This is very interesting point that we are going to study more carefully in our following experiments in the next paper. However, in the present research paper, we have observed that in the absence of centrosome, the number of TNT has been drastically reduced.

l.151.  “TNTs can account for an additional process to limit the detrimental consequence of centrosome amplification and to protect cells from genomic instability and transformation” This is an interesting question. However, this transfer reduced the centrosomes within a cell, but increased the number of centrosomes in other cells. Thus, one cell avoids centrosome amplification, but other cells will have multiple centrosomes. I do not understand the advantage for the cell community. Please, explain.

We believe that the transfer (or donation) of extra-centrosome from cancer cells to the healthy cells can contribute to acquisition of higher aggressive, invasive and metastatic phenotype.

l.186 “these exciting findings fit a model in which TNT serves as critical pathway to transfer extra-centrosome from malignant cells and alter healthy cells in the tumor environment”. Is it the answer to my previous question?

Exactly.

l.159. “in line with the failure of proper DNA segregation, we have also noticed in some case, the transfer of micronuclei through TNTs” I see in Fig 4 small spots at the basis of the cytoplasmic extensions, not through them. I think that this finding is not enough to affirm that micronuclei can move through TNT,

Yes, the reviewer is right. We consider to use the micronuclei marker such as Lamin and Lap2 to further validate these results in our following experiments.

Fig. 4A. Alpha-tubulin staining is poor. Any way the localization of pericentrin is evident.

Fig. 4B. The shape of the nucleus and tubulin staining are different in low magnification and detail.  I do not understand the arrow pointing the elongated yellow projections: is it gamma-tubulin? If it is so, this finding is very unusual. Please explain.

Concerning the first issue, the reviewer is right but this is because the images have been captured at different z-plane in different (please see the images below). This explanation has been added in the legend.

And about the second issue, according to the literature, Gamma-tubulin has been found not only in centrosome as a MTOC but also in the cytoplasm, and in association with cellular membranes (Yuba-Kubo et al., 2005; Hehnly & Doxsey, 2014; Draberova et al., 2017). Interestingly, it has been shown that in aggressive cancer cells gamma tubulin has more dispersive subcellular localization (Cho et al., 2010). Please see the supplemental figure for you in the annexed word of this response.

l.237. anti-ki67. Is it mono or polyclonal?

The gamma-tubulin antibody is lacking from materials and methods section. Again, if the gamma-tubulin antibody is from Sigma (as usual), how the AA can detect both alfa and gamma tubulin? I think that this is a polyclonal antibody.

We apologize for our carelessness and thank the reviewer for pointing this out. We have completed the information. Anti-ki67 is a polyclonal antibody and gamma-tubulin is a monoclonal antibody from Abcam.

Reviewer 3 Report

In the manuscript entitled “Centrosome, the newly identified passenger through Tunneling  Nanotube, increases binucleation and proliferation marker in receiving cells” Dubois and colleagues provide an interesting study about how the relationship between centrosome and TNT in normal and cancer cultured cells. According to the authors, centrosome localizes and orientates towards TNT protrusion site. Additionally authors claim to unveil a novel role for TNTs as a critical pathway serving to transfer extra-centrosome to protect malignant cells against aberrant centrosome amplification as well as to contribute to alter cells in the tumor environment

The importance of the work lies in the identification of the centrosome as a regulator of the TNT as well as a user of them, which could potentially serve as a druggable target. Most of the experimental work has been performed with HBEC-3, as well as a set of lung epithelial cell lines and tumor cells lines in presence of siRNA treatments when is needed. This study has novelty and addresses important questions on the regulation of cytoskeleton structures, however, there are concerns about presentation and/or interpretation of the data with a subset of experiments. The study remains non-conclusive and descriptive at some points. This evaluator considers that some more work needs to be done to present the results that they claim in a more solid way. This evaluator consider that some experiments need additional control conditions to help with the interpretation of the data and support the statements claimed by the authors.

On figure 1 authors used Centrinone (LCR-263), a PLK4 inhibitor, to induce centrosome depletion. Inhibitor also arrests cells a senescence-like G1 state, however, no data is shown demonstrating that the inhibitor is working at the indicated time and concentration. A cell cycle analysis will easily show the effect of the inhibitor.

On figure 2, authors showed that centrosome and Golgi localize close to the TNT protrusion site to facilitate directional intercellular exchanges of endocytic-recycling vesicles through TNTs. However, no mechanistical data is shown or discussed regarding these observations. Will it be possible to speculate about the molecular mechanism that underlies this effect?

On figure 3, authors used siRNA to deplete RASSF1A, a known regulator of centrosome number and TNT formation. However, authors did not show the depletion level achieved by the treatment (via WB and/or qPCR). Authors show only 2 images of the indirect effects of RASSF1A silencing trying to demonstrate the efficacy of RASSF1A depletion (supplementary 3). This evaluator consider that additional tests (mainly WB and/or qPCR) should be performed to demonstrate the efficacy of the siRNA treatment. On figure 3B authors claims that 70% of the RASSF1A-depleted HBEC-3 cells contained extra-centrosome and TNTs however no images of those cells are shown.

On figure 4A and B, authors used siRNA to deplete RASSF1A in HBEC-3 cells and promote the transference of the extra-centrosome to cells transfected with control siRNA after performing co-culture experiments. Although interesting, the experiment used cells transfected with RASSF1A as donors and cells transfected with control siRNA as acceptors, meaning that no controls are used on the donor population. Donor population should be transfected with specific and control siRNAs to demonstrate that the effect is treatment specific. In addition, the images are not very clear. It is not easy to identify the cells. Will it be possible to label the cells indicating the donor and the acceptor cells? Additionally, authors did not exclude the possibility that secretion of extracellular vesicles could underlie the transference observed.

In additional experiments, authors used “non-treated HBEC-3-Cherry cells or RASSF1A-depleted HBEC- 3Cherry cells were (co-) transfected with GFP-centrin plasmid and then co-cultured with either siNeg-transfected HBEC-3 or non-treated A549 cell lines for 24 hours” and identify that control HBEC-3 and/or A549 cells received GFP-centrin from RASSF1A without being transfected. However, authors do not use GFP-plasmid control. Authors did not exclude the possibility that control cells received genetic materials such as mRNA or the GFP-centrin plasmid from a donor cell or from the media rather than the protein. In lines 175-177 authors claim “We assumed that using TNT is the unique conduit for intercellular transfer of centrosome (identified by GFP-centrin) from donor HBEC-3Cherry cells into receiving cells (without cherry signal).”, however they do not support this statement with data.

Authors also claim that the presence of GFP-centrin in normal cells caused a significant increase of proliferation identified by Ki67 staining during co-culture experiments, however, 1) this observation was not done in donor cells. Did donor cells promote proliferation in other cells but not on themselves? 2) Authors do not explore other possibilities like paracrine signaling from RASSF1A depleted cells towards controls cells. Experiments using conditioned media may address this issue.

In lines 185-187 “Altogether, theses exciting findings fit a model in which TNT serves as critical pathway to transfer extra-centrosome from malignant cells and alter healthy cells in the tumor environment.” however they do not support this statement with data. The experiments are not done in a tumor microenvironment.

Minor points:

How was GFP-Centrin transfected? Using lipoiMAX? It will be necessary to indicate concentrations and/or amount or siRNA and plasmid transfected as well as the number of cells.

It will be interesting to indicate number of TNT/cells upon different conditions.

Author Response

We would like to thank the reviewer #3 for his/her interests and valuable comments on the manuscript.

On figure 1 authors used Centrinone (LCR-263), a PLK4 inhibitor, to induce centrosome depletion. Inhibitor also arrests cells a senescence-like G1 state, however, no data is shown demonstrating that the inhibitor is working at the indicated time and concentration. A cell cycle analysis will easily show the effect of the inhibitor.

We apologize for our carelessness and thank the reviewer for pointing this out. The pericentrin labeling was missing on the image, which has been added. These results shows that centrosome number has been reduced by almost 80% after seven days of centrinone treatment.

On figure 2, authors showed that centrosome and Golgi localize close to the TNT protrusion site to facilitate directional intercellular exchanges of endocytic-recycling vesicles through TNTs. However, no mechanistical data is shown or discussed regarding these observations. Will it be possible to speculate about the molecular mechanism that underlies this effect?

We thank the reviewer for this suggestion. It would have been interesting to explore this aspect. However, this would not be possible because it is beyond the scope of this paper. The main focus of these results is about the role of centrosome as principal microtubule nucleation/anchoring center and the well-known role of microtubule in directed trafficking (Rivero et al., 2009 ; Rios, 2014; Conduit et al., 2015 ; Sanders and Kaverina, 2015). Although, the research on TNTs uncovers a significant diversity of transport processes. In this regards, we would like to explore some hypothesis in our future experiments :

  • One observed and suspected transport mechanism is the exchange of vesicles by molecular motors. Does centrosome regulation of HDAC6/MT acetylation (Jiang et al., 2008; Ran et al., 2015) impact vesicle trafficking through TNTs by regulating kinesin function ?
  • It has been previously shown that Rab11 and its regulatory proteins localize to the centrosome. Additionally, evidence supports a role for vesicles containing Rab11, along with the PCM protein, γ-tubulin, and the motor protein, dynein, as potential vesicular carriers of proteins to the mitotic spindle poles (Hehnly & Doxsey, 2014). In this regard and in agreement with our previous data (Dubois et al., 2018), we would like to study in more details the role of Rab11 in vesicle trafficking through TNTs.
  • It might also be possible, that there are transport mechanisms which are cytoskeleton independent such as Cdc42 signaling molecule, which regulates both centrosome repositioning (Obino et al., 2016) and TNT-1 formation (Hanna et al., 2017).

On figure 3, authors used siRNA to deplete RASSF1A, a known regulator of centrosome number and TNT formation. However, authors did not show the depletion level achieved by the treatment (via WB and/or qPCR). Authors show only 2 images of the indirect effects of RASSF1A silencing trying to demonstrate the efficacy of RASSF1A depletion (supplementary 3). This evaluator consider that additional tests (mainly WB and/or qPCR) should be performed to demonstrate the efficacy of the siRNA treatment. On figure 3B authors claims that 70% of the RASSF1A-depleted HBEC-3 cells contained extra-centrosome and TNTs however no images of those cells are shown.

Concerning the first issue, we thank the reviewer for raising this critical issue. According to the reviewer’s suggestion, we have added a qPCR experiment to the supplementary figure 3 to show efficiency of RASSF1A depletion.

Although, in regards to the reviewer’s concern on figure 3B, we think there may have been some confusion here. The same image (Figure 3B) show that the three cells with TNT contain more than one centrosome. However, it is worth mentioning that as centrosome is very small (2micron) and there is only one or two of these structure in each cell, it is very difficult to take a picture which show the centrosome in all the cells and even more when there is two at different height within cytoplasm and/or TNT of each cells.

On figure 4A and B, authors used siRNA to deplete RASSF1A in HBEC-3 cells and promote the transference of the extra-centrosome to cells transfected with control siRNA after performing co-culture experiments. Although interesting, the experiment used cells transfected with RASSF1A as donors and cells transfected with control siRNA as acceptors, meaning that no controls are used on the donor population. Donor population should be transfected with specific and control siRNAs to demonstrate that the effect is treatment specific. In addition, the images are not very clear. It is not easy to identify the cells. Will it be possible to label the cells indicating the donor and the acceptor cells? Additionally, authors did not exclude the possibility that secretion of extracellular vesicles could underlie the transference observed.

While we appreciate the reviewer’s feedback, we respectfully disagree. TNTs allow the bidirectional-transport of many cellular components. Here, we show for the first time the transfer of extra-centrosome from the cells that contain more than one (RASSF1A depleted HBEC3-cells) to the cells with normal centrosome number (siNeg transfected HBEC3-cells). However, it is important to note that this co-culture experiment could account for control experiment as well because we could also observe the transfer from control cells with normal centrosome number (siNeg transfected HBEC3-cells) to the cells that contain more than one (RASSF1A depleted HBEC3-cells), which is not the case.

RASSF1A depletion was performed in labeled HBEC-3Cherry cell line, while the normal HBEC-3 cell line was used for siNeg transfection. To distinguish both cell populations and to help the readers for better visualization, we have separated cherry channel.

And we assume that using TNT is the unique conduit for intercellular transfer of centrosome because according to the literature, except in the case of cell division, the centrosome transfer from cell to cell, was never observed in any other context.

In additional experiments, authors used “non-treated HBEC-3-Cherry cells or RASSF1A-depleted HBEC- 3Cherry cells were (co-) transfected with GFP-centrin plasmid and then co-cultured with either siNeg-transfected HBEC-3 or non-treated A549 cell lines for 24 hours” and identify that control HBEC-3 and/or A549 cells received GFP-centrin from RASSF1A without being transfected. However, authors do not use GFP-plasmid control. Authors did not exclude the possibility that control cells received genetic materials such as mRNA or the GFP-centrin plasmid from a donor cell or from the media rather than the protein. In lines 175-177 authors claim “We assumed that using TNT is the unique conduit for intercellular transfer of centrosome (identified by GFP-centrin) from donor HBEC-3Cherry cells into receiving cells (without cherry signal).”, however they do not support this statement with data.

We thank the reviewer for pointing this limitation out. Although we agree that this is an important consideration, it is beyond the scope in this manuscript because we have focused on following the physical displacement of centrosome (GFP-centrin construct) through TNT, which is not appropriate and/ or possible through the transfection of GFP construct by itself, which doesn’t reveal any structure in the cells beside the whole cytoplasm. As discussed in previous question and demonstrated in Movie, we assume that using TNT is the unique conduit for intercellular transfer of centrosome because according to the literature, except in the case of cell division, the centrosome transfer from cell to cell, was never observed in any other context.

Authors also claim that the presence of GFP-centrin in normal cells caused a significant increase of proliferation identified by Ki67 staining during co-culture experiments, however, 1) this observation was not done in donor cells. Did donor cells promote proliferation in other cells but not on themselves? 2) Authors do not explore other possibilities like paracrine signaling from RASSF1A depleted cells towards controls cells. Experiments using conditioned media may address this issue.

In order to respond to this interesting question, it should be noted that according to the literature and our previous data, RASSF1A depletion disrupts successful progression and completion of cytokinesis, resulting in an increased incidence of binucleate cells as well as abscission failure during cytokinesis (Song et al., 2004; Guo et al., 2007; Keller et al., 2018). Therefore it is not surprising that upon receiving extra-centrosome, the control cells (containing wild type RASSF1A) showed an increase in Ki67 staining as a marker of proliferation.

Additionally, as the significant increase of Ki67 staining in control cells was concomitant with the presence of GFP-centrin, when it was compared to the control cell that have not received GFP-centrin, it suggest that transfer of extra-centrosome via TNT was responsible for this effect.

In lines 185-187 “Altogether, theses exciting findings fit a model in which TNT serves as critical pathway to transfer extra-centrosome from malignant cells and alter healthy cells in the tumor environment.” however they do not support this statement with data. The experiments are not done in a tumor microenvironment.

We understand the reviewer’s assessment and concern. However, we doesn’t imply that this is exactly what happen in tumor environment but the data emerged from this study only predict this idea. Additionally, the immunohistochemical experiment that has been realized on the paraffin-embedded tissue sections of the patients with NSCLC also support the functional significance of the centrosome transfer in a tumor microenvironment.

Minor points:

How was GFP-Centrin transfected? Using lipoiMAX? It will be necessary to indicate concentrations and/or amount or siRNA and plasmid transfected as well as the number of cells.

We apologize for our forgetfulness and thank the reviewer for pointing this out. We have completed this section of Materials and method in the revised manuscript.

It will be interesting to indicate number of TNT/cells upon different conditions.

We thank the reviewer for pointing this limitation out. Although we agree that this is an important suggestion, it is beyond the scope in this manuscript.

Round 2

Reviewer 1 Report

Author's Notes

Please see reviewer’s comments in red.

We would like to thank the reviewer #1 for his/her interests and valuable comments on the manuscript. As suggested, our article has undergone English language editing by MDPI. The text has been checked for correct use of grammar and common technical terms, and edited to a level suitable for reporting research in a scholarly journal.

General comments :

  1. The authors called the structures they are looking at TNTs.  These are not TNTs as described by the field.  TNTs are F-actin tubular structures that allow for the intercellular transfer of a number of organelles and signals.  The structures described in this article are much larger and contain microtubules, thus should not be referred to simply as TNTs but more specifically as microtubule-containing TNTs or MT-TNTs (for references see Wand and Gerdes, Cell death and differentiation, vol 22, 2015 or more recently, Han and Wang, IJMS vol. 22 2021).  The different types of TNT like structures should be briefly explained in the introduction.

As requested, the characteristic of TNTs have been explained in the introduction. According the characterization established by Bénard and co-workers, in 2015, there are two classes of TNT: 

  • TNT type 1 which are thicker and contain not only actin but also microtubules and intermediate filaments and 
  • TNT type 2 which are the thin ones and contain only actin filaments. 

The data in the literature suggest that these two types of TNT could have different functions, as large material (e.g., lysosomes, mitochondria) can only travel between cells via TNT type 1 on microtubules (Onfelt et al., 2006; Bénard et al., 2015; Wang & Gerdes, 2015). For these reasons and to pursue with functional experiments, we have focused only on type 1 TNT. 

As requested by reviewer and to avoid any confusion for the reader, we have modified the text of our manuscript and now specify “TNTs” when talking about all TNTs, and, TNTs-1 when talking about TNTs type 1.

These changes have improved the introduction as to the nature of the structures analyzed in this paper.  As a side note, it is a little surprising that the authors have not cited the original Rustom 2004 paper.  It is also unsettling that Benard et al., decided to call TNT1 the TNTs with both actin and tubulin and TNT2 actin only since most people in the field call TNTs the structures with actin only and MT-TNTs other structures.  While this might lead to some confusion in the field in the future, it is of course totally up to the authors to decide who they want to cite.  Since this is a short report it might explains the limited background information.

  1. The authors appear to use different words such as “polarizes towards”,  “displacements”, “displacement through”, “road to evacuate” and “transfer” interchangeably.  However these words are not all synonyms and should not be used appropriately.

We thank the reviewer for pointing this out. As requested, we tried to homogenize the use of these words.

The authors decided on using “displacement” however, they used it whether it meant “presence of X within TNT-1” or “moving within TNT-1” or “transferred between cells via TNT-1”.  Therefore, it is still misleading.  The authors demonstrated that the centrosomes are present and can move within TNT-1 but did not demonstrate that it is actively transferred from one cell to another.  Therefore, simply changing these different words to “displacement” within the text without additional experiments to show actual transfer from one cell to another is not valid.  In fact, the paper’s title has not been changed to mirror the actual data shown in the paper.

The authors have not shown conclusively that centrosomes are in fact Transferred.  The message of the paper and title cannot be that strong with the evidence provided.

3)   Overall, the evidence for actual transfer of centrosomes through TNTs is not conclusive.  While it is clear that the centrosomes are “polarize toward” or are “displaced” and can be found at the base or within MT-TNTs, the authors did not conclusively demonstrated transfer.

We thank the reviewer for raising this critical issue. According to the reviewer’s suggestion, we have replaced the word “ transfer” with either “displacement” or “displace”.

As stated above, the fact that the authors simply replaced transfer by displacement without changing the meaning behind it is meaningless.

The authors must be clear.  If you show the presence and movement within TNT-1, then say this.  This is what was clearly shown in this paper.  The authors did a good job showing this but they did not show transfer.

4)   In addition, there is no control for the possibility of centrosome transfer other than by TNTs either.  For instance in l. 175, the authors stated “ We assume that using TNT is the unique conduit for intercellular transfer of centrosome…’. Why not actually test this assumption by adding a microtubule destabilizing drug or centrinone to reduce TNT formation with the co-cultured cells to show that GFP-pericentrin is no longer observed in the receiving cells?  

We thank the reviewer for raising this important topic. We assume that using TNT is the unique conduit for intercellular transfer of centrosome because according to the literature, except in the case of cell division, the centrosome transfer from cell to cell, was never observed in any other context. However, to respond to the reviewer’s concern, we can’t realize these experiments due to technical impossibility. GFP-centrin transfection and co-culture experiments are realized after 24h but the centrinone treatment is effective after 7 days. Additionally, the Trypsination of treated cells with either centrinone or microtubule destabilizing drug cause the death of majority of cells prior to co-culture.

1) The red highlighted text is problematic.  The authors cannot assume that TNT is the only way simply because it was not shown any other way.  Scientists must look at the validity of their assumptions using experimental data not assumption that it is the case simply because others have not shown it happened before.  The authors are supposed to show that TNT-1 are the conduits for centrosome transfer.

2) The request in the first round of review is not impossible using microtubule destabilizing drugs (which do not require 7 days of treatments like centrinone).

Since we do not expect a quick and massive transfer to occur immediately upon co-culture.  This is why the authors probably chose 24 hrs of co-culture to actually be able to quantify transfer.  Here, the authors can mix the cells first and let them adhere to the plate a couple of hours and then do the 24 hrs co-culture along with treatment with microtubule destabilizing drugs.  You would expect to see a clear difference in transfer.

Alternatively, the authors could do co-culture in conditions that block cell-to-cell and TNT formation (i.e. with a filter) but does not block extracellular vesicle movement.

Specific comments:

Fig. 1:

(D) 200 cells were quantified in how many independent experiments?

The detail of each experiment has been added in the legends. Briefly, during three individual experiments (n=3), we have quantified the number of TNTs-1 in approximately 200 cells based on the characteristic discussed earlier. 

Appropriate changes.  Thank you.

(F) how many independent experiments are used and how many cells were counted?

The detail of each experiment has been added in the legends. Briefly, during three individual experiments, we have quantified the number of TNTs-1 in at least 200 cells based on the characteristic discussed earlier. 

Appropriate changes.  Thank you.

Fig. 2:

(B) Quantification of 50 cells is low.  How many independent experiments were acquired?  50 cells should be the minimum for 1 independent experiment and a minimum of 3 independent experiments should be acquired.

We thank the reviewer for pointing this out. In our revised manuscript, we have quantified the number of Golgi localization in another 50 TNTs-1 formed cells. Data are represented as the mean ± SEM from at least three individual experiments.

Appropriate changes.  Thank you.

Fig. 3:

(B) Quantification of centrosome displacement through TNTs.  The authors show the “% of siRASSF1 cells with or without extra centrosome displacement through TNTs” .  Here there is an important control missing.  The authors need to add a “scramble RNA control” to show the effect of siRNA and the difference in the centrosome displacement in siRASSF1 cells.  In addition, similar to the previous figures, the authors need to add how many independent experiments were done and n=40 is once again very low.

We think there may have been some confusion here. The control cells don’t have any extra centrosome. In our revised manuscript, we have quantified the displacement of centrosome in another experiment to increase the number of cells. Data are represented as the mean ± SEM from at least three individual experiments.

I apologize if my statement was not clear.  I am not sure why the authors thought that I understood that the control would have more extra centrosomes.

What I meant to say here is that for siRNA experiments, their should always be a “scramble siRNA” or a “non-targeting control RNAi”.  The “normal cells” are not the right control.  I am not sure if it was a mistake while writing the paper because the authors wrote the control were the “normal HBEC-3” cells BUT in the Material and method they wrote that they use a “non-targeting control RNAi from Dharmacon.”  What is it the actual control used?  Again, “normal cells” or siRNA neg” cells implies no treatment at all, which is not the correct control.  You need to take away the possibility that what you are observing ids not from the siRNA treatment itself.  That’s why you must use either a scramble RNA or a non-targeting control RNAi. If this is what the authors did, then they must make it clear in the text.  If the authors really only used the normal cells then they used the wrong controls. 

Title: Displacement of centrosome through TNTs need to be changed.  The authors are using it again as transfer.

(E) and Movie 1:  This example does not show the transfer of centrosome through TNTs.  The movie/time lapse starts with the centrosome already in the tube and then show movement towards the top cell.  What is shown here is the movement of centrosome in the tube, not transfer.  In order to show transfer the centrosome would have to be shown moving from the bottom cell to the top cell.  This is not the case.  The centrosome could have come from the top cell, move ~ 10 um down within the tube and then go back toward the center of the cell it originally came from.

We thank the reviewer for raising this critical issue. According to the reviewer’s suggestion, we have replaced the word “ transfer” with “displacement” .

As stated previously, simply changing “transfer” to displacement” but using it with the same meaning is useless.

However, it is worth to mention that this experiment presents many technical difficulties and challenges:  TNTs are very fragile, sensitive to the light and transient structures with an average survival time ranging from a few to tens of minutes (Sowinski et al., 2008 Bukoreshtliev et al., 2009). In addition, the movement of GFP-centrin is very difficult to visualize due to numerous issues: 

  •  unlike other organelles, which may number in the thousands per cell, centrosome typically occur in only one or two copies.
  •  centrosomes are very small structures (2 microns) and they become out of focus very easily
  •  and because of their small size, the strength of fluorescence signal reduces quickly and irremediably.

I know the challenges that the authors are facing working with TNTs.  However, they can’t say “this is to difficult to do”, and we will just assume that this is what is happening.  Other groups have shown actual transfer.  You cannot say transfer or in this case “displacement” with its meaning being “transfer” without actually scientifically show that it is the case.

The authors are probably correct in their assumption but for actual publication you need to show it using experimental approaches that are repeatable.

An alternative is to “turn down” the results.  For example, the authors should really emphasize what they have shown (i.e., Fig 1 and 2).  And change the tone of the paper, use the correct wording )(i.e., is found within TNT-1 and appear to move within these structures” etc…).  While we were not able to directly show the actual transfer of centrosome from one cell to another…. Our data strongly suggest that TNT-1 could be a conduit for centrosome transfer.  More direct evidence will be necessary etc…..  That is why the title is also way too strong and should not emphasize transfer but more what was shown which is that the centrosome is in fact critical for TNT-1 formation….

Fig.4:

(A) The tubular structure from the mCherry cell doesn’t appear to touch the neg cells on the left where the other points to the 2 centrosomes in (ii).  It would help to have the phase images overlaid with the fluorescent images (like in Fig. 2 and 3). In addition, there appears to be an mCherry cells on top of the negative cell (right outside of the highlighted yellow box ).  It is very difficult to see the outline of that cell and where its limits are.  Could the Pericentrin come from that cell?  A phase image could help better visualizing each individual cell and the cellular protrusion.

The TNTs are considered not attached to the substrate as they hover freely in medium and we can observe the bodies of cells and the middle of TNTs in two different optical sections and not with the same focus through microscope. In this regard, TNTs are even capable of passing above the other attached cells. The reviewer is right, in this image, in the continuity of mcherry, we can observe the cell-body of the neg cells (please see MT-staining) which is without mcherry fluorophores. We apologize from reviewer because we don’t have the phase images corresponding.

1) I am not sure why the authors wrote the red highlighted part.  There is no question that this is the case and this was never an issue. 

2) The authors could have simply taken a new representative picture and acquire the phase image.

(C), (E) and (G): There is no information on how many independent experiments were acquired and how many cells were counted in order to generate these plots.  This is important in order to determine the significance of the data.

We apologize for our carelessness and thank the reviewer for pointing this out. We have added the details in the legend. Briefly, data are represented as the mean ± SEM from at least three individual experiments. n>200 cells per individual experiment. Statistical significance was calculated and p value are indicated by asterisk: *p<0.05.

Appropriate changes.  Thank you.

Sup. Fig 4:

In general, it is difficult to see the TNTs actually bound to the other cells.  In Sup. Fig 4, the TNTs observed appear not to connect to the “receiving cells”.  As stated for Fig. 4., it would help to have the phase images overlaid with the fluorescent images (like in Sup. Fig 5).  

The TNTs are fragile structures which may break during fixation. This is why in some images we cannot see the point of attachment to the other cells. But as requested by reviewer, we have replaced the word “transfer” with “displacement” to focus on the presence of centrosome through TNTs and avoid any confusion for the reader in these images. We apologize from reviewer because we don’t have the phase images corresponding.

The authors could have simply taken a new representative picture and acquire the phase image.

Sup. Fig. 4 and 5: 

These Figures do not show transfer but simply the presence of centrosomes within the long cellular protrusions.  Transfer means movement from cell A to B.  None of these still images show that.

We thank the reviewer for raising this critical issue. According to the reviewer’s suggestion, we have replaced the word “ transfer” with “displacement” .

As stated previously, this is meaningless since the authors did not change their meassage and still use it as meaning “transfer” between cells.

That’s why the title again is misleading since it means transfer through TNTs.

Sup. Fig 6:

It doesn’t show TNT between the transfected cell and the non-treated A549 cell.

The reviewer is right, there is no TNT between these cells, but we assume that the transfer has been occurred trough TNTs before the time of image acquisition as there is no others possibilities for centrosome to move from GFP transfected cells to control cells.

Again, you can’t assume that it was via TNTs.  You must show that it is.  Simply because transfer of centrosome has not been shown before certainly does not mean that there “is no other possibilities”.  Transfer of centrosome has not really been looked at before so why would there be evidence of it, especially since as the authors stated, because of the small number this is not an easy task to accomplish.

Overall, the changes made in regarding cell numbers, number of experiments etc… are very useful.

The main problem is the fact that the authors are “overreaching” in their conclusions and what the paper actually shows.  It is very interesting to see that centrosomes can move to TNTs and that they might be important for TNT formation.  This is what the authors have shown in this report.  The rest (i.e., transfer and possible reason for transfer) have not been properly determined.  Therefore, the authors need to re-write the report to actually precisely state what their data showed, what their data suggest and then open it up to what they think it means and future studies…

Minor notes:

TNTs were not changed to TNT-1 in figures.

Instead of saying “we assume” p3., you might want to say “we hypothesize that a close …”.

Author Response

Please see reviewer’s comments in red.

Please see our new answers in green and italic

We would like to thank the reviewer #1 for his/her interests and valuable comments on the manuscript. As suggested, our article has undergone English language editing by MDPI. The text has been checked for correct use of grammar and common technical terms, and edited to a level suitable for reporting research in a scholarly journal.

General comments :

  1. The authors called the structures they are looking at TNTs.  These are not TNTs as described by the field.  TNTs are F-actin tubular structures that allow for the intercellular transfer of a number of organelles and signals.  The structures described in this article are much larger and contain microtubules, thus should not be referred to simply as TNTs but more specifically as microtubule-containing TNTs or MT-TNTs (for references see Wand and Gerdes, Cell death and differentiation, vol 22, 2015 or more recently, Han and Wang, IJMS vol. 22 2021).  The different types of TNT like structures should be briefly explained in the introduction.

As requested, the characteristic of TNTs have been explained in the introduction. According the characterization established by Bénard and co-workers, in 2015, there are two classes of TNT: 

  • TNT type 1 which are thicker and contain not only actin but also microtubules and intermediate filaments and 
  • TNT type 2 which are the thin ones and contain only actin filaments. 

The data in the literature suggest that these two types of TNT could have different functions, as large material (e.g., lysosomes, mitochondria) can only travel between cells via TNT type 1 on microtubules (Onfelt et al., 2006; Bénard et al., 2015; Wang & Gerdes, 2015). For these reasons and to pursue with functional experiments, we have focused only on type 1 TNT. 

As requested by reviewer and to avoid any confusion for the reader, we have modified the text of our manuscript and now specify “TNTs” when talking about all TNTs, and, TNTs-1 when talking about TNTs type 1.

These changes have improved the introduction as to the nature of the structures analyzed in this paper.  As a side note, it is a little surprising that the authors have not cited the original Rustom 2004 paper.  It is also unsettling that Benard et al., decided to call TNT1 the TNTs with both actin and tubulin and TNT2 actin only since most people in the field call TNTs the structures with actin only and MT-TNTs other structures.  While this might lead to some confusion in the field in the future, it is of course totally up to the authors to decide who they want to cite.  Since this is a short report it might explains the limited background information.

We thank the reviewer for his/her comprehension and we understand his/her concern. For conciseness, our preferred approach is to be consistent with our previous paper in 2018 (doi: 10.1186/s12964-018-0276-4) where we have also used TNT-1 for tunneling nanotubes which contain both actin and microtubules. However, we agree with the reviewer’s concern and we have started our introduction by citation of the original Rustom paper as follows :

 “Since their discovery by Rustom and colleagues in 2004 [1], Tunneling nanotubes (TNTs) represent a direct mode of intercellular communication that are gaining widespread importance in our understanding of processes involved in health and disease[2-4].”

  1. The authors appear to use different words such as “polarizes towards”,  “displacements”, “displacement through”, “road to evacuate” and “transfer” interchangeably.  However these words are not all synonyms and should not be used appropriately.

We thank the reviewer for pointing this out. As requested, we tried to homogenize the use of these words.

The authors decided on using “displacement” however, they used it whether it meant “presence of X within TNT-1” or “moving within TNT-1” or “transferred between cells via TNT-1”.  Therefore, it is still misleading.  The authors demonstrated that the centrosomes are present and can move within TNT-1 but did not demonstrate that it is actively transferred from one cell to another.  Therefore, simply changing these different words to “displacement” within the text without additional experiments to show actual transfer from one cell to another is not valid.  In fact, the paper’s title has not been changed to mirror the actual data shown in the paper.

The authors have not shown conclusively that centrosomes are in fact Transferred.  The message of the paper and title cannot be that strong with the evidence provided.

We thank the reviewer for these details which helped us to better understand his concern about this critical issue. We change not only the word assume with hypothesize but also we add some detail to answer why we think that TNTs are the unique conduit for centrosome transfer as follows:

“187-194”: We hypothesize that using TNTs-1 is the unique conduit for intercellular transfer of centrosomes (identified by GFP-centrin) from donor HBEC-3Cherry cells into receiving cells (without cherry signal) since the average size of centrosome (in vertebrates) is 1.6 ± 0.5 μm2 [42-44]  and among communication tools, only TNT-1 reach a thicknesses of over 700 nm sufficient to transfer of whole centrosomes from donor cells into receiving cells [45-46]. Indeed, gap junctional pores range in size from 11 to 24 Å in diameter [47]and extracellular vesicles, from nanometer-size exosomes (30-100nm) to submicron-size microparticles (100-1000 nm) [48-49].”

In addition, to avoid any confusion for the reader and according to reviewer’s suggestions, we have now modified these sentences:

Abstract:

28-32: Furthermore, using immunohistochemistry and live imaging, we observed for the first time the movement of extra centrosome within TNTs. In this regard, we hypothesize a novel role for TNTs-1 as a critical pathway serving to displace extra centrosomes and potentially to either protect malignant cells against aberrant centrosome amplification or contribute to altering cells in the tumor environment.

Introduction:

70: movement of centrosome within

Results & discussion:

123: TNTs-1 as an unexpected road for extra centrosomes

151: Time-lapse imaging further confirmed the presence and movement of centrosome within TNTs-1 between interconnected cells (Figure 3E, Movie 1).

Figure 3: The presence and movement of centrosome within TNTs

156: of such localization within TNT

162: localization of centrosome within TNTs-1

173: localization of micronuclei

3)   Overall, the evidence for actual transfer of centrosomes through TNTs is not conclusive.  While it is clear that the centrosomes are “polarize toward” or are “displaced” and can be found at the base or within MT-TNTs, the authors did not conclusively demonstrated transfer.

We thank the reviewer for raising this critical issue. According to the reviewer’s suggestion, we have replaced the word “ transfer” with either “displacement” or “displace”.

As stated above, the fact that the authors simply replaced transfer by displacement without changing the meaning behind it is meaningless.

The authors must be clear.  If you show the presence and movement within TNT-1, then say this.  This is what was clearly shown in this paper.  The authors did a good job showing this but they did not show transfer.

Again, we understand the reviewer’s concern and we hope that the changes that have been made, as stated above, help the readers to avoid any misinterpretation.

4)   In addition, there is no control for the possibility of centrosome transfer other than by TNTs either.  For instance in l. 175, the authors stated “ We assume that using TNT is the unique conduit for intercellular transfer of centrosome…’. Why not actually test this assumption by adding a microtubule destabilizing drug or centrinone to reduce TNT formation with the co-cultured cells to show that GFP-pericentrin is no longer observed in the receiving cells?  

We thank the reviewer for raising this important topic. We assume that using TNT is the unique conduit for intercellular transfer of centrosome because according to the literature, except in the case of cell division, the centrosome transfer from cell to cell, was never observed in any other context. However, to respond to the reviewer’s concern, we can’t realize these experiments due to technical impossibility. GFP-centrin transfection and co-culture experiments are realized after 24h but the centrinone treatment is effective after 7 days. Additionally, the Trypsination of treated cells with either centrinone or microtubule destabilizing drug cause the death of majority of cells prior to co-culture.

1) The red highlighted text is problematic.  The authors cannot assume that TNT is the only way simply because it was not shown any other way.  Scientists must look at the validity of their assumptions using experimental data not assumption that it is the case simply because others have not shown it happened before.  The authors are supposed to show that TNT-1 are the conduits for centrosome transfer.

We apologize if our statement was not clear.  We can certainly see merit in the question the reviewer poses, namely how we assume that TNTs are the only and unique way for centrosome transfer. As stated above, cells continuously communicate with each other through wide range of communication networks namely:

  • Endocrine/paracrine/autocrine signaling, a communication way for chemical messengers (hormones, neurotransmitter, small hydrophilic molecules, ….)
  • transport through gap junctions and electrical coupling, however, gap junctions are fewer than 40 amino acids in size (Duffy & Wallace, 1987; Millhorn & Hökfelt, 1988) and gap junctional pores range in size from 11 to 24 Å in diameter (Weber et al., 2004)
  • extracellular vesicle secretion: the extracellular vesicles, which carry RNAs, proteins and lipids, are enclosed by a lipid bilayer and range from nanometer-size exosomes (30-100nm) to submicron-size microparticles (100-1000 nm) (Vestad et al., 2017; Kao et al., 2019).
  • and tunneling nanotubes, TNTs-1 can be especially characterized by their dimensions, which reach thicknesses of over 700 nm and their length can dynamically be regulated from a few microns to over 100 μm (Wang et al., 2012; Austefjord et al., 2014).

As the average size of centrosome matrix (in vertebrates) is 1.6 ± 0.5 μm2 (Piehl et al., 2004; Conduit et al., 2010; Schatten, 2008), we assumed that the active transport of centrosome’s structure between two separate cells could only be possible through TNTs. Such idea is consistent with the results illustrated in the Figure 4 of our manuscript: due to the increase in TNT-1 formation after RASSF1A depletion (doi: 10.1186/s12964-018-0276-4), the number of control cells that were co-cultured with RASSF1A-depleted HBEC-3Cherry cells and presented GFP-centrin signals was significantly higher than control cells co-cultured with non-treated HBEC-3Cherry cells).

To avoid any confusion for the reader and according to reviewer’s suggestions, we have changed the word “assume” by “hypothesize” as follows:

“187-194”: We hypothesize that using TNTs-1 is the unique conduit for intercellular transfer of centrosomes (identified by GFP-centrin) from donor HBEC-3Cherry cells into receiving cells (without cherry signal) since the average size of centrosome (in vertebrates) is 1.6 ± 0.5 μm2 [42-44]  and among communication tools, only TNT-1 reach a thicknesses of over 700 nm sufficient to transfer of whole centrosomes from donor cells into receiving cells [45-46]. Indeed, gap junctional pores range in size from 11 to 24 Å in diameter [47]and extracellular vesicles, from nanometer-size exosomes (30-100nm) to submicron-size microparticles (100-1000 nm) [48-49].”

2) The request in the first round of review is not impossible using microtubule destabilizing drugs (which do not require 7 days of treatments like centrinone).

Since we do not expect a quick and massive transfer to occur immediately upon co-culture.  This is why the authors probably chose 24 hrs of co-culture to actually be able to quantify transfer.  Here, the authors can mix the cells first and let them adhere to the plate a couple of hours and then do the 24 hrs co-culture along with treatment with microtubule destabilizing drugs.  You would expect to see a clear difference in transfer.

We think there may have been some confusion here. We chose 24h because of the efficiency of GFP-centrin transfection, which is realized prior to the co-culture. Therefore, we still face the technical impossibility as we can’t do transfection after co-culture along with microtubule destabilizing treatment.

Alternatively, the authors could do co-culture in conditions that block cell-to-cell and TNT formation (i.e. with a filter) but does not block extracellular vesicle movement.

Please see our response above. The extracellular vesicles range from 30 to 1000 nm in size. Consequently, they are not large enough to carry the whole centrosome’s structure within themselves.

Specific comments:

Fig. 1:

(D) 200 cells were quantified in how many independent experiments?

The detail of each experiment has been added in the legends. Briefly, during three individual experiments (n=3), we have quantified the number of TNTs-1 in approximately 200 cells based on the characteristic discussed earlier. 

Appropriate changes.  Thank you.

Thank you.

(F) how many independent experiments are used and how many cells were counted?

The detail of each experiment has been added in the legends. Briefly, during three individual experiments, we have quantified the number of TNTs-1 in at least 200 cells based on the characteristic discussed earlier. 

Appropriate changes.  Thank you.

Thank you.

Fig. 2:

(B) Quantification of 50 cells is low.  How many independent experiments were acquired?  50 cells should be the minimum for 1 independent experiment and a minimum of 3 independent experiments should be acquired.

We thank the reviewer for pointing this out. In our revised manuscript, we have quantified the number of Golgi localization in another 50 TNTs-1 formed cells. Data are represented as the mean ± SEM from at least three individual experiments.

Appropriate changes.  Thank you.

Thank you.

Fig. 3:

(B) Quantification of centrosome displacement through TNTs.  The authors show the “% of siRASSF1 cells with or without extra centrosome displacement through TNTs” .  Here there is an important control missing.  The authors need to add a “scramble RNA control” to show the effect of siRNA and the difference in the centrosome displacement in siRASSF1 cells.  In addition, similar to the previous figures, the authors need to add how many independent experiments were done and n=40 is once again very low.

We think there may have been some confusion here. The control cells don’t have any extra centrosome. In our revised manuscript, we have quantified the displacement of centrosome in another experiment to increase the number of cells. Data are represented as the mean ± SEM from at least three individual experiments.

I apologize if my statement was not clear.  I am not sure why the authors thought that I understood that the control would have more extra centrosomes.

What I meant to say here is that for siRNA experiments, their should always be a “scramble siRNA” or a “non-targeting control RNAi”.  The “normal cells” are not the right control.  I am not sure if it was a mistake while writing the paper because the authors wrote the control were the “normal HBEC-3” cells BUT in the Material and method they wrote that they use a “non-targeting control RNAi from Dharmacon.”  What is it the actual control used?  Again, “normal cells” or siRNA neg” cells implies no treatment at all, which is not the correct control.  You need to take away the possibility that what you are observing is not from the siRNA treatment itself.  That’s why you must use either a scramble RNA or a non-targeting control RNAi. If this is what the authors did, then they must make it clear in the text.  If the authors really only used the normal cells then they used the wrong controls. 

We apologize if these sentences are not clear. We used the term “normal” HBEC-3 to distinguish these cells from labeled HBEC-3Cherry cell line. By the word “siNeg”, we mean the cells transfected with non-targeting control RNAi. To avoid any confusion for the reader and according to reviewer’s suggestions, we have modified the M&M as follows:

“242: non-targeting control RNAi (siNeg) from Dharmacon.”

Title: Displacement of centrosome through TNTs need to be changed.  The authors are using it again as transfer.

According to the reviewer’s suggestion, we have replaced the title as follows : The presence and movement of centrosome within TNTs

(E) and Movie 1:  This example does not show the transfer of centrosome through TNTs.  The movie/time lapse starts with the centrosome already in the tube and then show movement towards the top cell.  What is shown here is the movement of centrosome in the tube, not transfer.  In order to show transfer the centrosome would have to be shown moving from the bottom cell to the top cell.  This is not the case.  The centrosome could have come from the top cell, move ~ 10 um down within the tube and then go back toward the center of the cell it originally came from.

We thank the reviewer for raising this critical issue. According to the reviewer’s suggestion, we have replaced the word “ transfer” with “displacement” .

As stated previously, simply changing “transfer” to displacement” but using it with the same meaning is useless.

We agree with reviewer’s concern. According to the reviewer’s suggestion, we have replaced the title as follows: “Time-lapse imaging of the movement of centrosome through TNT between interconnected cells”, which is exactly what is observed in this movie.

However, it is worth to mention that this experiment presents many technical difficulties and challenges:  TNTs are very fragile, sensitive to the light and transient structures with an average survival time ranging from a few to tens of minutes (Sowinski et al., 2008 Bukoreshtliev et al., 2009). In addition, the movement of GFP-centrin is very difficult to visualize due to numerous issues: 

  •  unlike other organelles, which may number in the thousands per cell, centrosome typically occur in only one or two copies.
  •  centrosomes are very small structures (2 microns) and they become out of focus very easily
  •  and because of their small size, the strength of fluorescence signal reduces quickly and irremediably.

I know the challenges that the authors are facing working with TNTs.  However, they can’t say “this is to difficult to do”, and we will just assume that this is what is happening.  Other groups have shown actual transfer.  You cannot say transfer or in this case “displacement” with its meaning being “transfer” without actually scientifically show that it is the case. The authors are probably correct in their assumption but for actual publication you need to show it using experimental approaches that are repeatable.

An alternative is to “turn down” the results.  For example, the authors should really emphasize what they have shown (i.e., Fig 1 and 2).  And change the tone of the paper, use the correct wording )(i.e., is found within TNT-1 and appear to move within these structures” etc…).  While we were not able to directly show the actual transfer of centrosome from one cell to another…. Our data strongly suggest that TNT-1 could be a conduit for centrosome transfer.  More direct evidence will be necessary etc…..  That is why the title is also way too strong and should not emphasize transfer but more what was shown which is that the centrosome is in fact critical for TNT-1 formation….

We understand the reviewer’s concern that the transfer of centrosomes is not demonstrated directly by these data but that we provide strong arguments consistent with this idea. As already underlying, due to the numerous difficulties that we have faced for imaging centrosome in live cells (as stated previously), the movie provided here is the best that we had obtained after many tries for this paper. However, it is worth mentioning that in the following complementary project, we plan to carry out the live cell imaging of both centrosomes and microtubules which probably move along each other as centrosomes nucleate microtubules. We hypothesize that this Live cell imaging would show the point we want to make here as centrosome doesn’t necessarily “use the microtubule to travel”.

In this revised manuscript, as suggested by reviewer’s, we have toning down our interpretation, as follow:

216-219: Furthermore, except for the case of dividing cells, our results provide the very first strong assumption of centrosome displacement from cell to cell even if we failed to show the actual transfer of centrosome from the donor cell to the recipient cell directly.

Fig.4:

(A) The tubular structure from the mCherry cell doesn’t appear to touch the neg cells on the left where the other points to the 2 centrosomes in (ii). It would help to have the phase images overlaid with the fluorescent images (like in Fig. 2 and 3). In addition, there appears to be an mCherry cells on top of the negative cell (right outside of the highlighted yellow box). It is very difficult to see the outline of that cell and where its limits are.  Could the Pericentrin come from that cell?  A phase image could help better visualizing each individual cell and the cellular protrusion.

The TNTs are considered not attached to the substrate as they hover freely in medium and we can observe the bodies of cells and the middle of TNTs in two different optical sections and not with the same focus through microscope. In this regard, TNTs are even capable of passing above the other attached cells. The reviewer is right, in this image, in the continuity of mcherry, we can observe the cell-body of the neg cells (please see MT-staining) which is without mcherry fluorophores. We apologize from reviewer because we don’t have the phase images corresponding.

1) I am not sure why the authors wrote the red highlighted part.  There is no question that this is the case and this was never an issue. 

2) The authors could have simply taken a new representative picture and acquire the phase image.

We understand the reviewer’s concern and we apologize from reviewer because we had not done phase contrast during these series of manipulations. Sincerely, these images has been selected after numerous acquisitions to represents the presence of centrosome (at the beginning, in the middle and at the end of TNTs) between two connected cells. Given the technical difficulty of the acquisition of such representatives images, we offer the reviewer another alternative:  we added a dashed line to show perimeter of the cells on the separate a-tubulin channel. We hope that the reviewer will find this proposition satisfactory.

(C), (E) and (G): There is no information on how many independent experiments were acquired and how many cells were counted in order to generate these plots.  This is important in order to determine the significance of the data.

We apologize for our carelessness and thank the reviewer for pointing this out. We have added the details in the legend. Briefly, data are represented as the mean ± SEM from at least three individual experiments. n>200 cells per individual experiment. Statistical significance was calculated and p value are indicated by asterisk: *p<0.05.

Appropriate changes.  Thank you.

Thank you.

Sup. Fig 4:

In general, it is difficult to see the TNTs actually bound to the other cells.  In Sup. Fig 4, the TNTs observed appear not to connect to the “receiving cells”.  As stated for Fig. 4., it would help to have the phase images overlaid with the fluorescent images (like in Sup. Fig 5).  

The TNTs are fragile structures which may break during fixation. This is why in some images we cannot see the point of attachment to the other cells. But as requested by reviewer, we have replaced the word “transfer” with “displacement” to focus on the presence of centrosome through TNTs and avoid any confusion for the reader in these images. We apologize from reviewer because we don’t have the phase images corresponding.

The authors could have simply taken a new representative picture and acquire the phase image.

As stated above, we added the dashed line to show perimeter of the cells on the separate a-tubulin channel. We hope that the reviewer will find our responses to their comments satisfactory.

Sup. Fig. 4 and 5: 

These Figures do not show transfer but simply the presence of centrosomes within the long cellular protrusions.  Transfer means movement from cell A to B.  None of these still images show that.

We thank the reviewer for raising this critical issue. According to the reviewer’s suggestion, we have replaced the word “ transfer” with “displacement” .

As stated previously, this is meaningless since the authors did not change their meassage and still use it as meaning “transfer” between cells.

That’s why the title again is misleading since it means transfer through TNTs.

We thank the reviewer for raising this critical issue. According to the reviewer’s suggestion, we have replaced the word “displacement” with “localization”.

Sup. Fig 6:

It doesn’t show TNT between the transfected cell and the non-treated A549 cell.

The reviewer is right, there is no TNT between these cells, but we assume that the transfer has been occurred trough TNTs before the time of image acquisition as there is no others possibilities for centrosome to move from GFP transfected cells to control cells.

Again, you can’t assume that it was via TNTs.  You must show that it is.  Simply because transfer of centrosome has not been shown before certainly does not mean that there “is no other possibilities”.  Transfer of centrosome has not really been looked at before so why would there be evidence of it, especially since as the authors stated, because of the small number this is not an easy task to accomplish. Overall, the changes made in regarding cell numbers, number of experiments etc… are very useful.

The main problem is the fact that the authors are “overreaching” in their conclusions and what the paper actually shows.  It is very interesting to see that centrosomes can move to TNTs and that they might be important for TNT formation.  This is what the authors have shown in this report.  The rest (i.e., transfer and possible reason for transfer) have not been properly determined.  Therefore, the authors need to re-write the report to actually precisely state what their data showed, what their data suggest and then open it up to what they think it means and future studies…

We would like to thank again the reviewer for this comment and for pointing out all these critical issues. We hope that our new explanations along with toning down our interpretation respond to reviewer’s concern.

Minor notes:

TNTs were not changed to TNT-1 in figures.

We apologize for our carelessness and thank the reviewer for pointing this out. We have modified the figures.

Instead of saying “we assume” p3., you might want to say “we hypothesize that a close …”.

We thank the reviewer for his/her help and suggestion. As stated above, we have replaced the word “assume” with “hypothesize”.

Reviewer 3 Report

The authors have addressed properly all my concerns and/or scientifically argued them. Thank you for taking into account my comments in the revised version of the manuscript. I have no additional comments.

Author Response

We thank the reviewer for his/her kindness and his/her constructive remarks which allowed the improvement of our article.

Round 3

Reviewer 1 Report

Thank you for answering all of my questions.